# Protective role of neuronal and lymphoid cannabinoid CB$_2$ receptors in neuropathic pain

David Cabañero[1,2], Angela Ramírez-López[1], Eva Drews[3], Anne Schmöle[3], David M Otte[3], Agnieszka Wawrzczak-Bargiela[4], Hector Huerga Encabo[5,6], Sami Kummer[1], Antonio Ferrer-Montiel[2], Ryszard Przewlocki[7], Andreas Zimmer[3], Rafael Maldonado[1,8]*

[1]Laboratory of Neuropharmacology, Department of Experimental and Health Sciences, Universitat Pompeu Fabra, Barcelona, Spain; [2]Institute of Research, Development and Innovation in Healthcare Biotechnology of Elche (IDiBE), Universidad Miguel Hernández de Elche, Alicante, Spain; [3]Institute of Molecular Psychiatry, University of Bonn, Bonn, Germany; [4]Department of Pharmacology, Laboratory of Pharmacology and Brain Biostructure, Maj Institute of Pharmacology, Polish Academy of Sciences, Krakow, Poland; [5]Immunology Unit, Department of Experimental and Health Sciences, Universitat Pompeu Fabra, Barcelona, Spain; [6]Haematopoietic Stem Cell Laboratory, The Francis Crick Institute, London, United Kingdom; [7]Department of Molecular Neuropharmacology, Institute of Pharmacology, Polish Academy of Sciences, Krakow, Poland; [8]IMIM (Hospital del Mar Medical Research Institute), Barcelona, Spain

*For correspondence:
rafael.maldonado@upf.edu

Competing interests: The authors declare that no competing interests exist.

**Abstract** Cannabinoid CB$_2$ receptor (CB$_2$) agonists are potential analgesics void of psychotropic effects. Peripheral immune cells, neurons and glia express CB$_2$; however, the involvement of CB$_2$ from these cells in neuropathic pain remains unresolved. We explored spontaneous neuropathic pain through on-demand self-administration of the selective CB$_2$ agonist JWH133 in wild-type and knockout mice lacking CB$_2$ in neurons, monocytes or constitutively. Operant self-administration reflected drug-taking to alleviate spontaneous pain, nociceptive and affective manifestations. While constitutive deletion of CB$_2$ disrupted JWH133-taking behavior, this behavior was not modified in monocyte-specific CB$_2$ knockouts and was increased in mice defective in neuronal CB$_2$ knockouts suggestive of increased spontaneous pain. Interestingly, CB$_2$-positive lymphocytes infiltrated the injured nerve and possible CB$_2$ transfer from immune cells to neurons was found. Lymphocyte CB$_2$ depletion also exacerbated JWH133 self-administration and inhibited antinociception. This work identifies a simultaneous activity of neuronal and lymphoid CB$_2$ that protects against spontaneous and evoked neuropathic pain.

## Introduction

Cannabinoid CB$_2$ receptor (CB$_2$) agonists show efficacy in animal models of chronic inflammatory and neuropathic pain, suggesting that they may be effective inhibitors of persistent pain in humans (*Bie et al., 2018*; *Maldonado et al., 2016*; *Shang and Tang, 2017*). However, many preclinical studies assess reflexive-defensive reactions to evoked nociceptive stimuli and fail to take into account spontaneous pain, one of the most prevalent symptoms of chronic pain conditions in humans (*Backonja and Stacey, 2004*; *Mogil et al., 2010*; *Rice et al., 2018*) that triggers coping responses such as analgesic consumption. As a consequence, conclusions drawn from animal models relying on

evoked nociception may not translate into efficient pharmacotherapy in humans (*Huang et al., 2019*; *Mogil, 2009*; *Percie du Sert and Rice, 2014*), which underlines the need to apply more sophisticated animal models with clear translational value. Operant paradigms in which animals voluntarily self-administer analgesic compounds can provide high translatability and also identify in the same experimental approach potential addictive properties of the drugs (*Mogil, 2009*; *Mogil et al., 2010*; *O'Connor et al., 2011*). In this line, a previous work using a $CB_2$ agonist, AM1241, showed drug-taking behavior in nerve-injured rats and not in sham-operated animals, suggesting spontaneous pain relief and lack of abuse potential of $CB_2$ agonists (*Gutierrez et al., 2011*), although the possible cell populations and mechanisms involved remain unknown. In addition, a recent multicenter study demonstrated off-target effects of this compound on anandamide reuptake, calcium channels and serotonin, histamine and kappa opioid receptors (*Soethoudt et al., 2017*).

$CB_2$, the main cannabinoid receptors in peripheral immune cells (*Fernández-Ruiz et al., 2007*; *Schmöle et al., 2015a*), are found in monocytes, macrophages and lymphocytes, and their expression increases in conditions of active inflammation (*Schmöle et al., 2015b*; *Shang and Tang, 2017*). The presence of $CB_2$ in the nervous system was thought to be restricted to microglia and limited to pathological conditions or intense neuronal activity (*Manzanares et al., 2018*). However, recent studies using electrophysiological approaches and tissue-specific genetic deletion revealed functional $CB_2$ also in neurons, where they modulate dopamine-related behaviors (*Zhang et al., 2014*) and basic neurotransmission (*Quraishi and Paladini, 2016*; *Stempel et al., 2016*). Remarkably, the specific contribution of immune and neuronal $CB_2$ to the development of chronic pathological pain has not yet been established.

This work investigates the participation of neuronal and non-neuronal cell populations expressing $CB_2$ in the development and control of chronic neuropathic pain. We used a pharmacogenetic strategy combining tissue-specific $CB_2$ deletion and drug self-administration to investigate spontaneous neuropathic pain. Constitutive and conditional knockouts lacking $CB_2$ in neurons or monocytes were nerve-injured, subjected to operant self-administration of the specific $CB_2$ agonist JWH133 (*Soethoudt et al., 2017*) and were evaluated for nociceptive and anxiety-like behavior. We also explored infiltration of $CB_2$-positive immune cells in the injured nerve of mice receiving bone marrow transplants from $CB_2$-GFP BAC mice. Finally, immunological blockade of lymphocyte extravasation was used to investigate the effect of this cell type on spontaneous neuropathic pain and its involvement on the pain-relieving effects of the cannabinoid $CB_2$ agonist.

## Results

### Self-administration of a $CB_2$ receptor agonist to alleviate spontaneous pain and anxiety-associated behavior

$CB_2$ agonists have shown efficacy reducing evoked sensitivity and responses of negative affect in mouse models of chronic pain (*Maldonado et al., 2016*). Although antinociception is a desirable characteristic for drugs targeting chronic neuropathic pain, it is unclear whether the pain-relieving effects of the $CB_2$ agonist would be sufficient to elicit drug-taking behavior in mice and the cell populations involved. To answer these questions, mice underwent a PSNL or a sham surgery and were placed in operant chambers where they had to nose poke on an active sensor to obtain i.v. self-administration of the $CB_2$ agonist JWH133 or vehicle (*Figure 1A*). Sham mice or nerve-injured animals receiving vehicle or the low dose of JWH133 (0.15 mg/kg/inf) did not show significant differences in active nose-poking during the last 3 days of the drug self-administration period (*Figure 1B*, *Figure 1—figure supplement 1A*). Conversely, nerve-injured mice exposed to the high dose of JWH133 (0.3 mg/kg/inf) showed higher active responses than sham-operated mice receiving the same treatment (*Figure 1B*, *Figure 1—figure supplement 1A*). As expected, the operant behavior of sham-operated mice exposed to JWH133 was not different from that of sham mice exposed to vehicle, suggesting absence of reinforcing effects of the $CB_2$ agonist in mice without pain (*Figure 1B*, *Figure 1—figure supplement 1A*). The number of nose pokes on the inactive sensor was similar among the groups, indicating absence of locomotor effects of the surgery or the pharmacological treatments. Thus, operant JWH133 self-administration was selectively associated to the neuropathic condition.

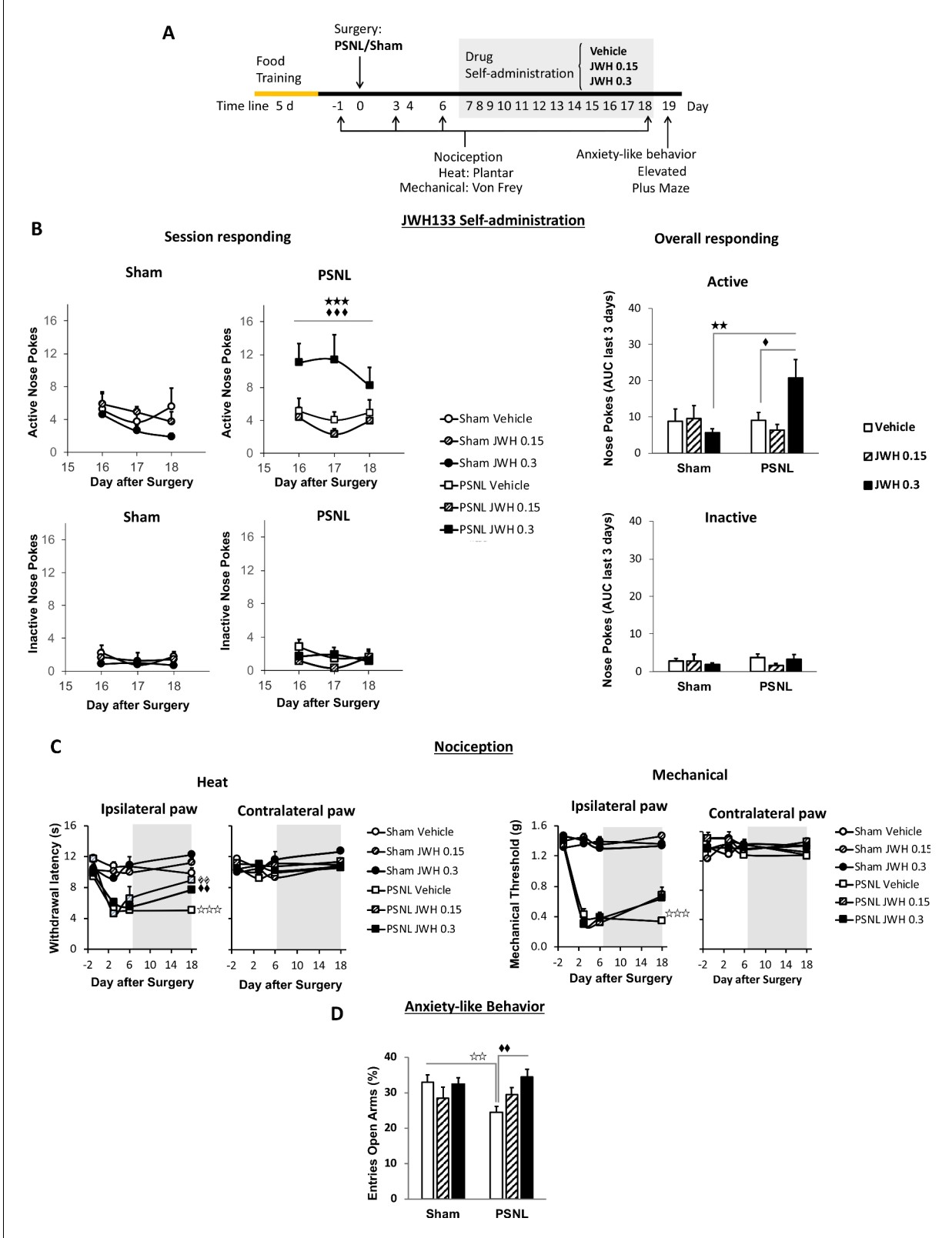

**Figure 1.** C57BL/6J mice self-administer a CB$_2$ receptor agonist with antinociceptive and anxiolytic-like properties. (**A**) Timeline of the drug self-administration paradigm. Mice were trained in Skinner boxes (5 days, 5d) where nose-poking an active sensor elicited delivery of food pellets. Partial sciatic nerve ligation (PSNL) or sham surgery were conducted (day 0) followed by jugular catheterization to allow intravenous (i.v.) drug infusion. From days 7 to 18, mice returned to the operant chambers and food was substituted by i.v. infusions of JWH133 (0.15 or 0.3 mg/kg/inf.). Mechanical and

*Figure 1 continued on next page*

*Figure 1 continued*

thermal sensitivity were assessed before (−1) and 3, 6 and 18 days after PSNL using Plantar and von Frey tests. Anxiety-like behavior was measured at the end (day 19) with the elevated plus maze. (B) Nerve-injured mice poked the active sensor to consume the high dose of JWH133 (0.3 mg/kg/inf.). (C) PSNL-induced ipsilateral thermal and mechanical sensitization (days 3 and 6). JWH133 inhibited thermal hypersensitivity but the effect on mechanical nociception was not significant (D) Nerve-injured mice receiving vehicle showed decreased percentage of entries to the open arms of the elevated plus maze, whereas PSNL mice receiving JWH133 0.3 mg/kg/inf. did not show this alteration. N = 5–10 mice per group. Shaded areas represent drug self-administration. Mean and error bars representing SEM are shown. Stars represent comparisons vs. sham; diamonds vs. vehicle. *p<0.05; **p<0.01; ***p<0.001.

The online version of this article includes the following source data and figure supplement(s) for figure 1:

**Source data 1.** JWH133 self-administration, antinociception and anxiolytic-like effects in C57BL6/J mice.
**Figure supplement 1.** JWH133 self-administration after nerve injury or sham surgery in C57BL6/J mice and food-maintained operant training before the drug self-administration.
**Figure supplement 1—source data 1.** Operant training and full JWH133 self-administration in C57BL6/J mice.

Nociceptive responses to thermal and mechanical stimuli were assessed before and after the self-administration period (days −1, 3, 6 and 18). Before the treatment with the $CB_2$ agonist, all nerve-injured mice developed heat and mechanical hypersensitivity in the ipsilateral paw (*Figure 1C*). After self-administration (shaded area, *Figure 1C*) mice exposed to JWH133 showed a significant reduction in heat hypersensitivity (*Figure 1C*, day 18, ipsilateral paw), although the alleviation of mechanical hypersensitivity did not reach statistical significance in this experiment. No significant drug effects were observed in the contralateral paws.

We also studied affective-like behavior in mice exposed to this chronic pain condition. Anxiety-like behavior was enhanced in nerve-injured mice treated with vehicle, as these mice visited less frequently the open arms of the elevated plus maze (*Figure 1D*). This emotional response was absent in nerve-injured mice repeatedly exposed to the high dose of JWH133 (*Figure 1D*). Therefore, the high dose of JWH133 elicited a drug-taking behavior selectively associated to spontaneous pain relief, and had efficacy limiting the pronociceptive effects of the nerve injury and its emotional-like consequences.

## $CB_2$ receptor mediates JWH133 effects on spontaneous pain alleviation

JWH133 has been recently recommended as a selective $CB_2$ agonist to study the role of $CB_2$ in biological and disease processes due to its high selectivity for this receptor (*Soethoudt et al., 2017*). To investigate the specificity of the $CB_2$ agonist in our model, the high dose of JWH133 (0.3 mg/kg/inf) was offered to nerve-injured mice constitutively lacking the $CB_2$ ($CB_2$ KO) and to C57BL/6J wild-type mice. $CB_2$ KO mice showed a significant disruption of JWH133-taking behavior on the last sessions of the drug self-administration period (*Figure 2A*, *Figure 2—figure supplement 1A*). Overall discrimination between the active and inactive sensors was also significantly blunted in $CB_2$ KO mice (Source Data File) and inactive nose pokes were similar in both groups of mice, indicating absence of genotype effect on locomotion (*Figure 2A*, *Figure 2—figure supplement 1A*). The disruption of drug-taking behavior shown in $CB_2$ KO mice was accompanied by an inhibition of JWH133 effects on nociceptive and affective behavior (*Figure 2B*, *Figure 2C*).

$CB_2$ KO and C57BL/6J mice developed similar thermal and mechanical hypersensitivity in the injured paw (*Figure 2B*, day 6, Ipsilateral paw), although $CB_2$ KO mice also developed hypersensitivity in the contralateral paw, as previously described (*Racz et al., 2008*). While C57BL/6J mice showed significant recovery of thermal and mechanical thresholds after JWH133 self-administration (*Figure 2B*, day 18), $CB_2$ KO mice showed no effects of the treatment on mechanical sensitivity (*Figure 2B*, day 18, Mechanical) and a partial recovery of the thresholds to heat stimulation (*Figure 2B*, day 18, Heat). Contralateral mechanical sensitization was still present in $CB_2$ KO mice exposed to the $CB_2$ agonist (*Figure 2B*, Contralateral paw). Likewise, nerve-injured C57BL/6J mice showed less anxiety-like behavior after JWH133 self-administration than $CB_2$ KO mice (*Figure 2C*), suggesting that these anxiolytic-like effects of JWH133 are mediated by $CB_2$. Hence, $CB_2$ KO mice showed reduced drug-taking behavior accompanied by blunted inhibition of JWH133 effects on mechanical nociception and anxiety-like behavior, confirming mediation of these effects by $CB_2$.

JWH133 has shown effects interacting with the Transient Receptor Potential Ankyrin1 (TRPA1) (*Soethoudt et al., 2017*), a receptor needed for thermal pain perception (*Vandewauw et al.,*

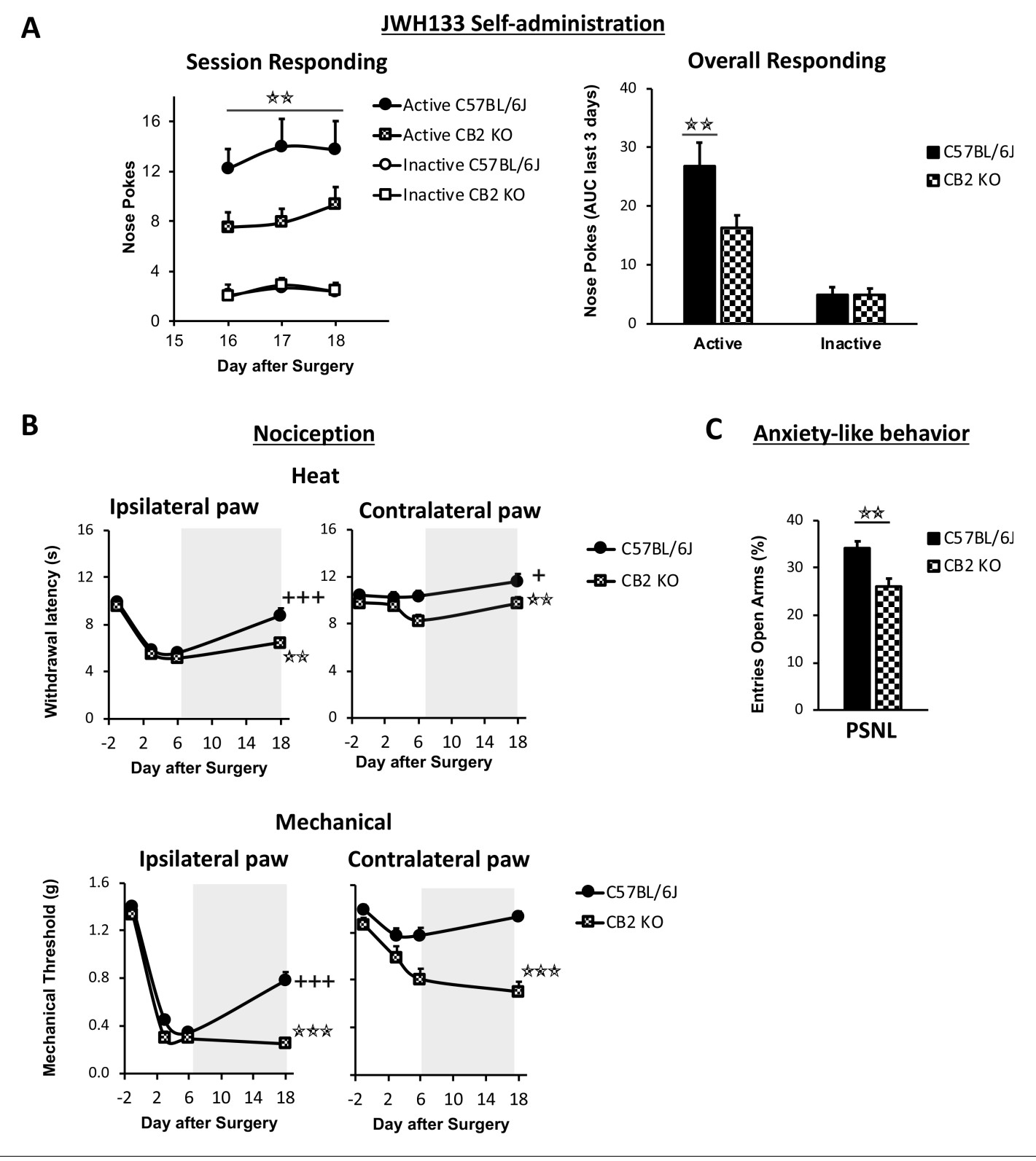

**Figure 2.** Nerve-injured mice constitutively lacking CB$_2$ receptor show disruption of JWH133 intake and blunted effects of the drug. CB$_2$ constitutive knockout mice (CB$_2$ KO) and C57BL/6J mice were food-trained in Skinner boxes (Food training, 5 days), subjected to a partial sciatic nerve ligation (PSNL, day 0), catheterized and exposed to high doses of the CB$_2$ agonist JWH133 (0.3 mg/kg/inf., days 7 to 18). Nociceptive sensitivity to heat (Plantar) and mechanical (von Frey) stimulation were measured before and after the nerve injury (−1,3,6,18), and anxiety-like behavior was evaluated at the end

*Figure 2 continued on next page*

*Figure 2 continued*

(day 19). (**A**) CB$_2$ KO mice showed decreased active operant responding for the CB$_2$ agonist. (**B**) The effects of JWH133 on thermal nociception were reduced in constitutive knockout mice. CB$_2$ KO mice showed contralateral mechanical and thermal sensitization and complete abolition of JWH133 effects on mechanical hypersensitivity. (**C**) Anxiety-like behavior after the treatment worsened in CB$_2$ KO mice. N = 16–19 mice per group. Mean and error bars representing SEM are shown. Shaded areas represent drug self-administration. Stars represent comparisons vs. C57BL/6J mice; crosses represent day effect. *p<0.05; **p<0.01; ***p<0.001.

The online version of this article includes the following source data and figure supplement(s) for figure 2:

**Source data 1.** JWH133 self-administration, antinociception and anxiolytic-like effects in nerve-injured CB$_2$ constitutive knockout mice.

**Figure supplement 1.** JWH133 self-administration in C57BL6/J and CB$_2$ constitutive knockout (CB2 KO) mice and food-maintained operant training before nerve injury and drug self-administration.

**Figure supplement 1—source data 1.** Operant training and full JWH133 self-administration in CB$_2$ constitutive knockout mice.

**Figure supplement 2.** Mice lacking TRPA1 receptor retain JWH133- effects.

**Figure supplement 2—source data 1.** Antinociceptive effect of JWH133 in TRPA1 knockout mice.

*2018*), that could also participate in other nociceptive responses. In order to assess a possible effect of the CB2 agonist on TRPA1 receptors in vivo, we conducted an additional experiment in which we compared the antinociceptive efficacy of JWH133 in sham and nerve-injured TRPA1 knockout mice (TRPA1 KO) and wild-type mice. After 7 days of the nerve injury, vehicle or i.p. doses of JWH133 (5 and 10 mg/kg) were administered to nerve-injured and sham-operated mice, and mechanical and heat nociception were assessed 30 and 75 min later, respectively. We observed similar effects of JWH133 inhibiting mechanical hypersensitivity in TRPA1 KO and WT mice (*Figure 2—figure supplement 2A*). Interestingly, TRPA1 KO mice showed a prominent inhibition of neuropathic thermal hypersensitivity (*Figure 2—figure supplement 2B*). In spite of this lack of sensitivity, a significant general effect was observed in nerve-injured mice with the high dose of JWH133 (10 mg/kg), regardless of the genotype of the mice. Thus, the results on mechanical sensitivity suggest that these effects are not due to an interaction of the drug with the TRPA1 receptor. The lack of thermal hypersensitivity observed in the TRPA1 KO mice may occlude possible JWH133 effects on neuropathic thermal hyperalgesia through TRPA1; however, a significant effect of JWH133 was observed in both strains after the nerve injury, suggesting that at least the CB2 receptor is involved in the inhibitory effect on thermal hyperalgesia.

## Participation of neuronal and monocyte CB$_2$ receptor in neuropathic pain symptomatology

CB$_2$ receptors were initially described in peripheral immune cells (*Munro et al., 1993*), although they have been found in multiple tissues including the nervous system. In order to distinguish the participation of CB$_2$ from different cell types on spontaneous neuropathic pain, we conducted the self-administration paradigm in nerve-injured mice lacking CB$_2$ in neurons (Syn-Cre+ mice) or in monocyte-derived cells (LysM-Cre+) and in their wild-type littermates (Cre Neg). Syn-Cre+ mice showed increased active operant responding for JWH133 (*Figure 3A*, *Figure 3—figure supplement 1A*), suggesting increased spontaneous pain and possible decrease of drug effects. On the other hand, LysM-Cre+ mice did not show significant alteration of drug-taking behavior (*Figure 3A*, *Figure 3—figure supplement 1A*). Inactive responding was also similar between Cre Neg and knockout mice. Thus, data from the drug self-administration experiments showed persistence of drug effects in the different genotypes and increased self-administration in mice lacking neuronal CB$_2$, suggestive of increased spontaneous pain.

We also measured antinociceptive and anxiolytic-like effects of JWH133 self-administration (*Figure 3B*, *Figure 3C*). The three mouse lines showed similar evoked responses to nociceptive stimulation after nerve injury (*Figure 3B*). A slight but significant impairment on the effect of JWH133 on mechanical sensitivity was found in Syn-Cre+ mice (*Figure 3C*) in spite of the increased JWH133 consumption, compatible with reduced efficacy of JWH133 in this mouse strain. The assessment of anxiety-like behavior did not reveal apparent differences among the three genotypes (*Figure 3C*). Thus, the increased JWH133 consumption observed in Syn-Cre+ mice was not reflected in increased anxiolysis and JWH133 antinociceptive effects were blunted, suggesting partial involvement of neuronal CB$_2$ in the development of spontaneous and evoked neuropathic pain. To investigate a possible involvement of peripheral neuronal CB$_2$ on the antinociceptive effects of JWH133, an additional

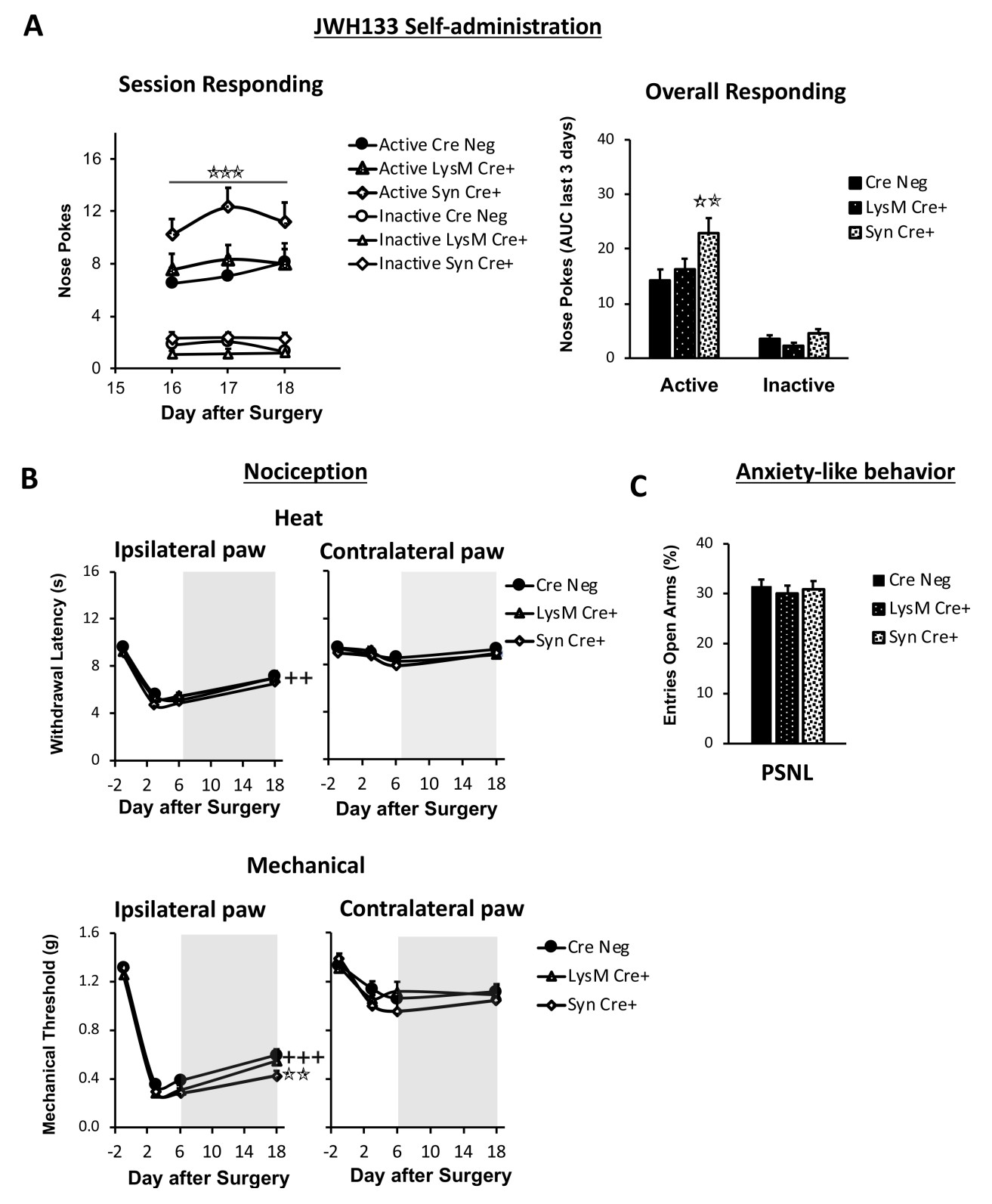

**Figure 3.** Nerve-injured mice defective in neuronal CB$_2$ receptor show increased self-administration of the CB$_2$ agonist JWH133 and a decrease in the antinociceptive effects of the drug. Mice lacking CB$_2$ in neurons (Syn-Cre+), in monocytes (LysM-Cre+) or their wild-type littermates (Cre Neg) were food-trained in Skinner boxes (Food training, 5 days), subjected to partial sciatic nerve ligation (PSNL, day 0), catheterized and exposed to JWH133 (0.3 mg/kg/inf., days 7 to 18). Nociceptive sensitivity to heat (Plantar) and mechanical (von Frey) stimulation were measured before and after nerve injury

*Figure 3 continued on next page*

*Figure 3 continued*

(−1,3,6,18), anxiety-like behavior was evaluated at the end (day 19). (**A**) Syn-Cre+ mice showed increased active operant responding for JWH133 in the last sessions of the self-administration period (**B**) All mouse strains showed decreased heat nociception after JWH133 treatment, and Syn-Cre+ mice showed reduced effects of JWH133 on mechanical nociception. (**C**) Every mouse strain showed similar anxiety-like behavior after JWH133 self-administration. No significant differences were found between LysM-Cre+ and Cre Neg mice. N = 18–36 mice per group. Mean and error bars representing SEM are shown. Shaded areas represent drug self-administration. Stars represent comparisons vs. Cre Neg mice; crosses represent day effect. *p<0.05; **p<0.01; ***p<0.001.

The online version of this article includes the following source data and figure supplement(s) for figure 3:

**Source data 1.** JWH133 self-administration, antinociception and anxiolytic-like effects in nerve-injured neuronal or microglial CB$_2$ knockout mice.

**Figure supplement 1.** JWH133 self-administration in mice lacking CB$_2$ in neurons or monocytes and their wild-type littermates and food-maintained operant training before nerve injury and drug self-administration.

**Figure supplement 1—source data 1.** Operant training and full JWH133 self-administration in neuronal or microglial CB$_2$ knockout mice.

**Figure supplement 2.** Mice lacking CB$_2$ in Nav1.8+ peripheral neurons show unaltered JWH133 antinociceptive effects.

**Figure supplement 2—source data 1.** Antinociceptive effect of JWH133 in CB$_2$ Nav1.8 Cre+ mice lacking CB2 in primary afferent neurons.

experiment was performed in floxed CB$_2$ mice expressing Cre recombinase in Nav1.8+ primary afferents (Nav1.8-Cre+, lacking CB$_2$ only in primary afferent nociceptive fibers, *Figure 3—figure supplement 2*), and in floxed littermates lacking Cre (Cre Neg). After 7 days of the nerve injury, vehicle or i.p. doses of JWH133 (5 and 10 mg/kg) were administered to nerve-injured and sham-operated mice, and mechanical and heat nociception were assessed 30 and 75 min later, respectively. No significant differences were observed between both genotypes in mechanical or thermal sensitivity (*Figure 3—figure supplement 2*) revealing that CB$_2$ primarily expressed in nociceptors were not involved in the antinociceptive effects of JWH133.

## Infiltration of non-neuronal CB$_2$ receptor-GFP+ cells in the injured nerve

The persistence of JWH133 effects after genetic deletion of CB$_2$ from neurons and monocyte-derived cells led us to hypothesize that CB$_2$ of other cell types may still exert neuromodulatory effects. To investigate possible infiltration of non-neuronal GFP+ cells in the injured nerve, we transplanted bone marrow cells from C57BL/6J or CB$_2$-GFP BAC mice to lethally irradiated CB57BL/6J-recipient mice (*Figure 4—figure supplement 1*). Mice transplanted with bone marrow from CB$_2$-GFP mice (CB$_2$ -GFP BMT) or from C57BL/6J mice (C57BL/6J BMT) were exposed to a partial sciatic nerve ligation or a sham surgery and dorsal root ganglia were collected 14 days later. A significant infiltration of non-neuronal GFP+ cells was revealed in nerve injured CB$_2$-GFP BMT mice (~30 cells/mm$^2$, *Figure 4A*, *Figure 4—figure supplement 2*), indicating that CB$_2$ -expressing cells invaded the injured nerve. Immunostaining to identify these cell types revealed co-localization with macrophage and lymphocyte markers. Nearly 60% of infiltrating macrophages and around 40% of the lymphocytes were found to be GFP+ (*Figure 4B*, *Figure 4C*, *Figure 4—figure supplement 3*). Surprisingly, a significant percentage of neurons was also found to express GFP in CB$_2$-GFP BMT mice (*Figure 4D*). The percentage of GFP+ neurons was higher in nerve-injured mice (~4% of total neurons) than in sham-operated animals (~2%, *Figure 4D*, *Figure 4—figure supplement 4*). Since GFP could only come from bone-marrow transplanted cells, this finding suggests a transfer of CB$_2$ from bone-marrow derived cells to neurons. Hence, nerve injury facilitated the invasion of affected ganglia by CB$_2$-positive immune cells and promoted a neuronal GFP expression compatible with transfer of CB$_2$ from immune cells to neurons.

## Lymphocyte involvement on JWH133 efficacy

The discovery of CB$_2$-expressing lymphocytes invading the dorsal root ganglia of nerve-injured mice prompted us to investigate the role of this cell type in spontaneous neuropathic pain. To answer this question, C57BL/6J mice were repeatedly treated with a control IgG or with an antibody targeting intercellular adhesion molecule 1 (ICAM1), a protein required for lymphocyte extravasation (*Labuz et al., 2009*). Mice under treatment with anti-ICAM1 or with the control IgG were exposed to JWH133 self-administration. Instead of reducing the intake of the CB$_2$ agonist, anti-ICAM1 significantly increased active nose poking to obtain i.v. JWH133 without altering the inactive nose poking (*Figure 5A*, *Figure 5—figure supplement 1A*), suggesting increased spontaneous pain. This result is in agreement with previous works showing protection against chronic inflammatory and

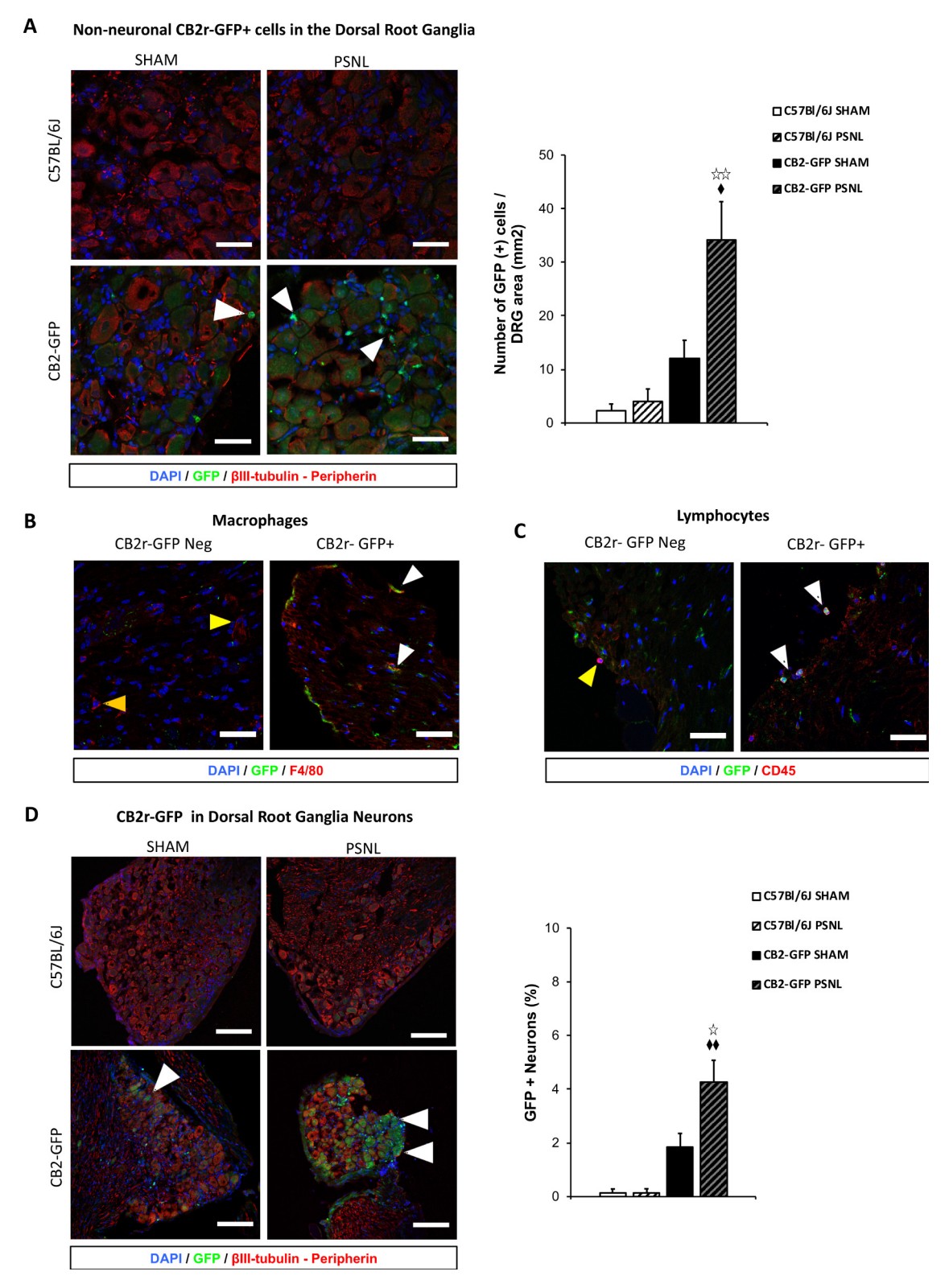

**Figure 4.** $CB_2$ receptor-GFP immune cells infiltrate the dorsal root ganglia of the injured nerve and GFP from bone-marrow-derived cells is also found inside peripheral neurons. The figure shows images of L3-L5 dorsal root ganglia from sham (SHAM) or nerve-injured mice (PSNL) transplanted with bone marrow cells from $CB_2$ GFP BAC mice ($CB_2$-GFP) or C57BL6/J mice (C57BL6/J). (A, D) Dorsal root ganglia sections stained with the nuclear marker DAPI (Blue), anti-GFP (Green), and neuronal markers anti-β-III tubulin and anti-peripherin (Red). (A) $CB_2$-GFP mice showed significant infiltration of GFP

*Figure 4 continued on next page*

*Figure 4 continued*

+ bone-marrow-derived cells after the nerve injury, whereas sham or nerve-injured C57BL6/J mice did not show significant GFP immunorreactivity. Split channels in *Figure 4—figure supplement 2*. (B) Co-localization of CB$_2$-GFP and the macrophage marker anti-F4/80. Co-staining with anti-GFP and anti-F4/80 revealed GFP+ (~60%) and GFP negative macrophages infiltrating the injured nerve. Split channels in *Figure 4—figure supplement 3A*. (C) Co-staining with anti-GFP and anti-CD45 revealed GFP+ (~40%) and GFP-negative lymphocytes infiltrating the injured nerve. Split channels in *Figure 4—figure supplement 3B*. (D) CB$_2$-GFP mice showed a percentage of GFP+ neurons that was enhanced with the nerve injury. Scale bar, 140 μm. Split channels in *Figure 4—figure supplement 4*. Scale bar for B), C), D), 45 μm. Yellow arrows point to GFP negative cells and white arrows to GFP+ cells. A certain degree of image processing has been applied equally across the entire merged images for optimal visualization. N = 2–3 mice per group. Means and error bars representing SEM are shown. Stars represent comparisons vs. sham; diamonds vs. C57BL6/J. *p<0.05, **p<0.01, ***p<0.001. Flow cytometry of blood from CB$_2$-GFP and C57BL6/J mice in *Figure 4—figure supplement 1*. Additional images of Sham and nerve-injured CB$_2$-GFP mice in *Figure 4—figure supplements 5* and *6*. Specificity tests for Tyramide Signal Amplification in *Figure 4—figure supplement 7*. Controls for antibody specificity in *Figure 4—figure supplement 8*.

The online version of this article includes the following source data and figure supplement(s) for figure 4:

**Source data 1.** CB$_2$ GFP cells in dorsal root ganglia of C57BL6/J nerve-injured mice after bone-marrow transplants from CB$_2$ GFP BAC mice.
**Figure supplement 1.** Bone marrow transplantation from CB$_2$ GFP BAC to C57BL6/J mice yields mice with peripheral blood cells expressing GFP.
**Figure supplement 1—source data 1.** Flow cytometry of blood from C57BL6/J mice transplanted with bone marrow from CB$_2$ GFP BAC mice.
**Figure supplement 2.** Non-neuronal CB$_2$ -GFP+ cells in the Dorsal Root Ganglia.
**Figure supplement 3.** Presence of CB$_2$ receptor-GFP in immune cells in the Dorsal Root Ganglia.
**Figure supplement 4.** CB$_2$-GFP in Dorsal Root Ganglia Neurons.
**Figure supplement 5.** Additional images of CB$_2$-GFP in Dorsal Root Ganglia Neurons.
**Figure supplement 6.** Higher magnification of merged images shown in *Figure 4—figure supplement 5*.
**Figure supplement 7.** Tyramide signal amplification (TSA) for optimal visualization of GFP in the Dorsal Root Ganglia.
**Figure supplement 8.** Controls for antibody specificity in the Dorsal Root Ganglia.

neuropathic pain mediated by lymphoid cells (*Labuz et al., 2009*; *Baddack-Werncke et al., 2017*). Interestingly, thermal and mechanical nociception before self-administration were similar in anti-ICAM1 and control IgG-treated mice (*Figure 5B*). After self-administration, the alleviation of thermal sensitivity was similar in control IgG and anti-ICAM1-treated mice (*Figure 5B*), but mice treated with anti-ICAM1 also showed an abolition of the antinociceptive effect of JWH133 on mechanical sensitivity (*Figure 5B*). This was evident in spite of the increased drug-taking behavior shown by mice treated with anti-ICAM1 (*Figure 5A*), which reveals decreased antinociceptive efficacy of JWH133 in these mice. On the contrary, anxiety-like behavior was similar in Control IgG and anti-ICAM1 mice (*Figure 5C*). To confirm an effect of the antibody treatment on lymphocyte infiltration, RT-PCR for white blood cell markers was performed in the dorsal root ganglia of mice subjected to the behavioral paradigm. As expected, a significant decrease in T cell markers CD2 and CD4 was observed in mice treated with anti ICAM-1 (*Figure 5D,T* cell panel). Interestingly, anti ICAM-1 also showed a pronounced increase in B cell marker CD19 (*Figure 5D*) and no alteration of the macrophage marker C1q (*Figure 5D*). Hence, our results reveal that lymphoid cells are involved in spontaneous neuropathic pain and are also necessary for the antinociceptive effect of JWH133 on mechanical sensitivity.

## Discussion

This work shows a protective function of CB$_2$ from neurons and lymphocytes on spontaneous neuropathic pain and the involvement of these cell populations in CB$_2$-induced antinociception, as revealed by increased self-administration of the CB$_2$ agonist JWH133 in mice defective in lymphocyte and neuronal CB$_2$. Previous works already demonstrated antinociceptive and emotional-like effects of CB$_2$ agonists in rodent models of acute and chronic pain (*Gutierrez et al., 2011*; *Ibrahim et al., 2003*; *Jafari et al., 2007*; *La Porta et al., 2015*; *Maldonado et al., 2016*). Our results provide evidence that the effect of the CB$_2$ agonist is sufficient to promote drug-taking behavior in nerve-injured mice for alleviation of spontaneous pain, but it is void of reinforcing effects in animals without pain, suggesting the absence of abuse liability. This absence of reinforcement adds value to the modulation of pain through CB$_2$ agonists, since current available agents for neuropathic pain treatment have reduced efficacy and often show addictive properties in humans and rodents (*Attal and Bouhassira, 2015*; *Bonnet and Scherbaum, 2017*; *Bura et al., 2018*; *Finnerup et al., 2015*; *Hipólito et al., 2015*; *O'Connor et al., 2011*).

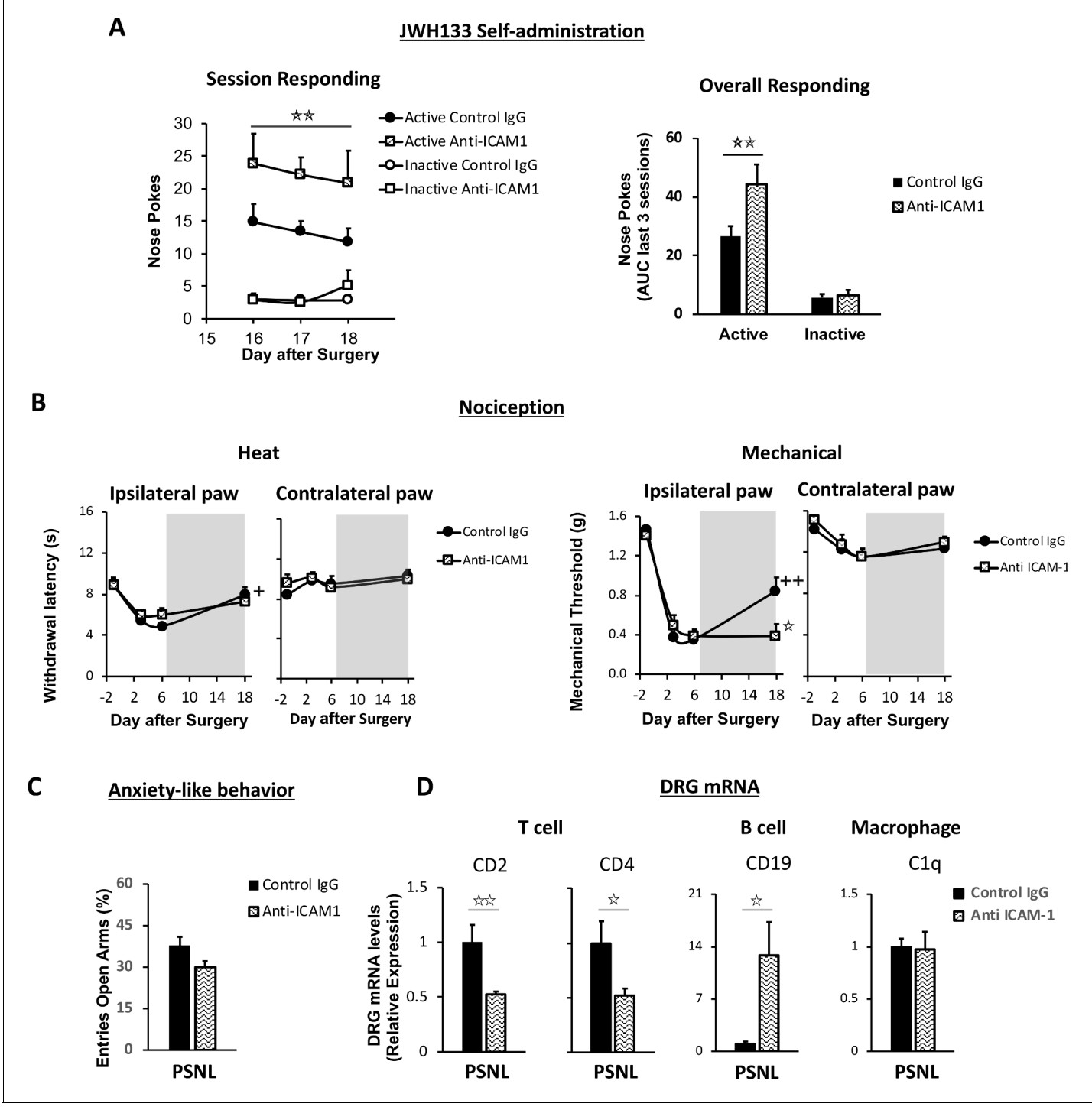

**Figure 5.** Lymphocytes modulate the effects of JWH133 on spontaneous pain and mechanical nociception. C57BL/6J mice were food-trained in Skinner boxes (Food training, 5 days), subjected to partial sciatic nerve ligation (PSNL, day 0), catheterized and exposed to high doses of the CB$_2$ agonist JWH133 (0.3 mg/kg/inf., days 7 to 18). Treatments with Anti-ICAM1 (an antibody that inhibits lymphocyte extravasation) or control IgG were given intraperitoneally once a day from day 0 until the end of self-administration. Nociceptive sensitivity to heat (Plantar) and mechanical (von Frey) stimulation was measured before and after nerve injury (−1,3,6,18), and anxiety-like behavior was evaluated at the end (day 19). Dorsal root ganglia were collected for mRNA analysis (**A**) Mice treated with anti-ICAM1 showed increased active responding for JWH133. (**B**) Thermal nociception after JWH133 self-administration was similar in mice treated with anti-ICAM1 or control IgG. Conversely, JWH133 effects on mechanical nociception were abolished by anti-ICAM1. (**C**) Anxiety-like behavior was similar in anti-ICAM1 and control IgG mice. (**D**) Levels of mRNA from T cell markers CD2 and CD4 were decreased in the dorsal root ganglia of anti-ICAM1 mice. Conversely, levels of B cell marker CD19 increased. Macrophage marker C1q was

*Figure 5 continued on next page*

*Figure 5 continued*

unaffected. N = 6–7 mice per group. Shaded areas represent drug self-administration. Mean and error bars representing SEM are shown. Stars represent comparisons vs. control IgG group; crosses indicate day effect. *p<0.05; **p<0.01; ***p<0.001.

The online version of this article includes the following source data and figure supplement(s) for figure 5:

**Source data 1.** JWH133 self-administration, antinociception and anxiolytic-like effects in nerve-injured C57BL6/J mice treated with anti-ICAM1.

**Figure supplement 1.** JWH133 self-administration of nerve-injured mice treated with anti-ICAM1 or control IgG and food-maintained operant training before nerve injury and drug self-administration.

**Figure supplement 1—source data 1.** Operant training and full JWH133 self-administration in nerve-injured C57BL6/J mice treated with anti-ICAM1.

A previous work using the $CB_2$ agonist AM1241 showed drug-taking behavior and antinociception in nerve-injured rats (*Gutierrez et al., 2011*), although a recent multicenter study demonstrated off-target effects of this drug (*Soethoudt et al., 2017*). The disruption of JWH133 effects observed in constitutive knockout mice confirms that the relief of spontaneous pain and the effects reducing mechanical nociception and anxiety-like behavior are mediated by $CB_2$ stimulation. However, the $CB_2$ agonist partially preserved its effects promoting drug self-administration and relieving thermal hypersensitivity in CB2KO mice, suggesting that JWH133 may also act through other receptors. Our results with the nerve-injured TRPA1 knockout mice revealed that JWH133 preserves its efficacy inducing antinociception in the absence of this receptor. Therefore, TRPA1 does not seem to play a relevant role in the antinociceptive response induced by JWH133 in our model of neuropathic pain mice. Interestingly, TRPA1 deletion prevented the development of thermal hyperalgesia after the nerve injury in our study and formalin (0.5%) evoked nocifensive behaviors are also lost in the TRPA1 knockout (*McNamara et al., 2007*). Another possibility is a minor involvement of $CB_1$ receptor in JWH133 self-administration after the nerve injury, since this compound is a selective $CB_2$ agonist that exhibited 40-fold higher affinity for mouse $CB_2$ than for $CB_1$ receptor (*Soethoudt et al., 2017*). Our experiments were not designed to rule out this possible minor participation. However, the inactive responding in the operant self-administration sessions was similar after JWH133 or vehicle in nerve-injured mice, indicating an absence of primary $CB_1$-related side effects such as motor alteration that are classical effects observed in mice after the administration of $CB_1$ agonists (*Rodríguez de Fonseca et al., 1998*). Previous works using higher equivalent doses of a $CB_2$ agonist with similar potency (*Soethoudt et al., 2017*) showed also absence of motor impairment in nerve-injured rats (*Gutierrez et al., 2011*). Since JWH133 selectivity for human $CB_2$ vs. human $CB_1$ is higher (153-fold selectivity), we do not foresee any concern with CB1-related behavioral effects using similar molecules in humans.

Nerve-injured mice defective in neuronal $CB_2$ showed higher JWH133 intake than wild-type littermates, indicating persistence of drug effects and increased spontaneous pain when neurons do not express $CB_2$. Thus, increased self-administration suggests an enhanced affective-motivational component of pain and not reduced drug efficacy on this aspect, since nerve-injured C57BL6/J mice exposed to the low JWH133 dose did not show compensatory increased self–administration (*Figure 1*). Importantly, mechanical and thermal neuropathic hypersensitivity before drug self-administration were similar in neuronal knockouts and their wild-type littermates, which suggests different mechanisms of spontaneous pain and evoked nociception. In addition, mechanical nociception measured after JWH133 was more severe in neuronal $CB_2$ knockouts than in wild-type littermates, which indicates decreased JWH133 efficacy on mechanical antinociception. Several studies described the presence of $CB_2$ mRNA and functional $CB_2$ receptor in neuronal populations from different areas of the rain (*Stempel et al., 2016*; *Zhang et al., 2014*). However, other works using targeted expression of fluorescent proteins under the control of the mouse gene *Cnr2* failed to describe $CB_2$ expression in neurons (*López et al., 2018*; *Schmöle et al., 2015a*). Our results agree with a role of neuronal $CB_2$ during painful neuroinflammatory conditions, a setting that was not studied before in mice defective in neuronal $CB_2$. Although we cannot provide a precise localization of the neurons involved in the increased spontaneous and evoked pain of neuronal knockout mice, the similar JWH133 response of $CB_2$ Nav1.8 Cre+ mice lacking $CB_2$ in primary afferent neurons (Nav1.8-Cre+) and wild-type mice could indicate involvement of a different set of neurons or increased relevance of $CB_2$ from immune sources. Thermal hypersensitivity and anxiety-like behavior measured after self-administration was similar in neuronal knockouts and wild-type mice, which indicates involvement of non-

neuronal cell populations. However, it should also be considered that the neuronal knockout mice had higher JWH133 consumption. Thus, a possible lack of efficacy could also be present for thermal antinociception and inhibition of anxiety-like behavior. Although a neuronal involvement was found, $CB_2$ neuronal knockouts did not recapitulate the phenotype of mice constitutively lacking $CB_2$, suggesting additional cell types involved in the effects of $CB_2$agonists.

We investigated the effects of JWH133 promoting its own consumption and inducing antinociception and anxiolysis in $CB_2$ LysM-Cre+ mice, mainly lacking $CB_2$ in monocytes, the precursors of microglial cells. We did not observe a microglial participation in these pain-related phenotypes, which may be due to an incomplete deletion of $CB_2$ in microglia through LysM-driven Cre expression (*Blank and Prinz, 2016*). Previous studies in mice constitutively lacking $CB_2$ described an exacerbated spinal cord microgliosis after nerve injury (*Nozaki et al., 2018*; *Racz et al., 2008*), which suggested a relevant role of $CB_2$ controlling glial reactivity. Since spinal microgliosis participates in the increased pain sensitivity after a neuropathic insult and macrophages and microglia express $CB_2$, blunted effects of JWH133 were expected in microglial $CB_2$ knockouts. However, monocyte-derived cells did not seem to be involved in the analgesic effects mediated by the exogenous activation of $CB_2$ in these experimental conditions.

The immunohistochemical analysis of dorsal root ganglia from mice transplanted with bone marrow cells of $CB_2$GFP BAC mice (*Schmöle et al., 2015a*) revealed a pronounced infiltration of immune cells expressing $CB_2$ in the dorsal root ganglia after nerve injury. Macrophages and lymphocytes expressing $CB_2$ were found at a time point in which nerve-injured mice present mechanical and thermal hypersensitivity and self-administer compounds with demonstrated analgesic efficacy (*Bura et al., 2013*; *Bura et al., 2018*). Interestingly, GFP expression was also found in neurons, suggesting a transfer of $CB_2$ from peripheral immune cells to neurons. An explanation for this finding may come from processes of cellular fusion or transfer of cargo between peripheral blood cells and neurons (*Alvarez-Dolado et al., 2003*; *Ridder et al., 2014*). Bone-marrow-derived cells fuse with different cell types in a process of cellular repair that increases after tissue damage. These events may be particularly important for the survival of neurons with complex structures that would otherwise be impossible to replace (*Giordano-Santini et al., 2016*). Alternatively, extracellular vesicles drive intercellular transport between immune cells and neurons (*Budnik et al., 2016*). Earlier studies showed incidence of fusion events between bone-marrow-derived cells and peripheral neurons in a model of diabetic neuropathy (*Terashima et al., 2005*), and similar processes were observed in central neurons after peripheral inflammation (*Giordano-Santini et al., 2016*; *Ridder et al., 2014*). Functional contribution of these mechanisms to neuronal $CB_2$ expression has not yet been explored, although cargo transfer between immune cells and neurons could modify neuronal functionality and it could offer novel therapeutic approaches to modulate neuronal responses (*Budnik et al., 2016*). Hence, $CB_2$ coming from white blood cells and present in neurons could be significant modulators of spontaneous neuropathic pain and may be contributors to the analgesic effect of $CB_2$ agonists.

Our results suggest participation of lymphoid cells on spontaneous neuropathic pain, but not on basal neuropathic hypersensitivity, highlighting possible differences on the pathophysiology of these nociceptive manifestations. In addition, lymphoid cells were essential for the effects of JWH133 alleviating mechanical sensitivity after a nerve injury. These hypotheses were evaluated by using anti-ICAM1 antibodies that impair lymphocyte extravasation. Previous studies revealed that anti-ICAM1 treatment inhibited opioid-induced antinociception in a model of neuropathic pain (*Celik et al., 2016*; *Labuz et al., 2009*). According to the authors, stimulation of opioid receptors from the immune cells infiltrating the injured nerve evoked the release of opioid peptides that attenuated mechanical hypersensitivity. Lymphocyte $CB_2$ could also be involved in the release of leukocyte-derived pain-modulating molecules. Experiments assessing the function of ICAM1 (*Celik et al., 2016*; *Deane et al., 2012*; *Labuz et al., 2009*) showed the participation of this protein on lymphocyte extravasation. In agreement, we observed a decrease of T cell markers in the dorsal root ganglia of mice receiving anti-ICAM1. Anti-ICAM1 treatment also increased the mRNA levels of the B cell marker CD19 in the dorsal root ganglia. Since ICAM1-interacting T cells show activity limiting B cell populations (*Deane et al., 2012*; *Zhao et al., 2006*), it is likely that the absence of T cells in the nervous tissue increased infiltration of B lymphocytes. B cells are involved in the severity of neuroinflammatory processes and have been linked to pain hypersensitivity (*Huang et al., 2016*; *Jiang et al., 2016*; *Li et al., 2014*; *Zhang et al., 2016*). Interestingly, $CB_2$ receptors restrict glucose and energy supply of B cells (*Chan et al., 2017*), which may alter their cytokine production as

previously described for macrophages and T cells. However, the participation of $CB_2$ from B cells on neuropathic pain has not yet been established. Our results indicate an increase in JWH133 consumption that could be driven by an increased infiltration of B cells. Overall, the results with the ICAM-1 experiment suggest a relevant participation of lymphoid $CB_2$ on painful neuroinflammatory responses. Our data also underscore the interest of investigating the role of lymphoid cells in brain regions involved in pain, anxiety or negative reinforcement during chronic neuroinflammatory processes.

While we identify $CB_2$-expressing neurons and lymphocytes as cellular entities involved in spontaneous and evoked neuropathic pain, the efficacy of the $CB_2$ agonist eliciting its own self-administration to alleviate pain was only disrupted in constitutive $CB_2$ knockout mice. These results indicate that the cell types involved in the negative reinforcement induced by JWH133 were suboptimally targeted in our experimental conditions, probably because different cell types expressing $CB_2$ are involved in this phenotype. Vascular cells may represent alternative participants of this behavior since JWH133 showed local vasodilatory effects (*McDougall et al., 2008*) and endothelial functional $CB_2$ receptor was found in cerebral microvasculature (*Ramirez et al., 2012*; *Onaivi et al., 2012*).

In summary, the contribution of neurons and lymphocytes to the effects of $CB_2$ agonists on spontaneous and evoked pain suggests a coordinated response of both cell types after the nerve injury. $CB_2$-expressing lymphocytes could participate in pain sensitization through release of pain-related molecules and the observed responses are also compatible with transfer of $CB_2$ between immune cells and neurons. Hence, bone-marrow-derived cells may provide a source of functional $CB_2$ that was not considered before and could clarify the controversial presence of these receptors in neurons. Nociceptive and affective manifestations of chronic neuropathic pain are therefore orchestrated through neuronal and immune sites expressing $CB_2$, highlighting the functional relevance of this cannabinoid receptor in different cell populations.

Our results on operant JWH133 self-administration depict $CB_2$ agonists as candidate analgesics for neuropathic conditions, void of reinforcing effects in the absence of pain. These pain-relieving effects involve the participation of $CB_2$ from neurons and lymphocytes preventing the neuroinflammatory processes leading to neuropathic pain. Therefore, $CB_2$ agonists would be of interest for preventing neuropathic pain development and the potential trials to evaluate this effect should consider starting $CB_2$ agonist treatment before or shortly after the induction of neuropathic insults, as in our study, in contrast to the treatment strategies used in previous clinical trials. The identification of a cannabinoid agonist simultaneously targeting the behavioral traits and the multiple cell types involved in the pathophysiology of chronic neuropathic pain acquires special relevance in a moment in which the absence of efficient analgesics void of abuse liability has become a major burden for public health.

# Materials and methods

**Key resources table**

| Reagent type (species) or resource | Designation | Source or reference | Identifiers | Additional information |
|---|---|---|---|---|
| Strain, strain background (*Mus musculus*, male) | C57BL/6J | Charles Rivers, France | RRID:IMSR_JAX:000664 | |
| Genetic reagent (*M. musculus*) | $CB_2$ KO | Institute of Molecular Psychiatry, University of Bonn, Germany | RRID:MGI:2663848 | *Buckley et al., 2000* PMID:10822068 (male) |
| Genetic reagent (*M. musculus*) | SynCre+-$Cnr2^{fl/fl}$:: $Cnr2^{fl/fl}$ | Institute of Molecular Psychiatry, University of Bonn, Germany | | (C57BL/6J background, male) |
| Genetic reagent (*M. musculus*) | LysMCre+-$Cnr2^{fl/fl}$:: $Cnr2^{fl/fl}$ | Institute of Molecular Psychiatry, University of Bonn, Germany | | (C57BL/6J background, male) |
| Genetic reagent (*M. musculus*) | Nav1.8Cre+-$Cnr2^{fl/fl}$:: $Cnr2^{fl/fl}$ | Institute of Molecular Psychiatry, University of Bonn, Germany | | (C57BL/6J background, male) |

*Continued on next page*

Continued

| Reagent type (species) or resource | Designation | Source or reference | Identifiers | Additional information |
|---|---|---|---|---|
| Genetic reagent (*M. musculus*) | TRPA1 KO | Universidad Miguel Hernández, Spain | RRID:MGI:3625358 | *Kwan et al., 2006*. PMID:16630838 (male) |
| Strain, strain background (*M. musculus,* male) | C57BL/6JRccHsd | Universidad Miguel Hernández, Spain | | Envigo |
| Antibody | Anti-mouse ICAM-1 (Hamster, monoclonal, clone 3e2) | BD Biosciences, USA | 550287 | (150 µg/day i.p.) |
| Antibody | IgG from rabbit serum (Unconjugated) | Sigma-Aldrich, Germany | I5006 | (150 µg/day i.p.) |
| Antibody | Allophycocyanin-conjugated anti-mouse CD11b (Monoclonal) | eBioscience, USA | cn.17–0112 | Flow cytometry (1:300) |
| Antibody | Phycoerythrin-conjugated anti-mouse B220 (Monoclonal) | eBioscience, USA | cn.12–0452 | Flow cytometry (1:100) |
| Antibody | Phycoerythrin/cyanine-conjugated anti-mouse CD3 (Monoclonal) | BioLegend, USA | cn.100320 | Flow cytometry (1:100) |
| Antibody | Rabbit anti-peripherin (Polyclonal) | Thermo Fisher, USA | PA3-16723 | IHC (1:200) |
| Antibody | Rabbit anti-GFP antibody (Polyclonal) | Thermo Fisher, USA | A11122 | IHC (1:2000) |
| Antibody | Rabbit anti-β-III tubulin (Polyclonal) | Abcam, UK | Ab18207 | IHC (1:1000) |
| Antibody | Rat anti-CD45R/B220 APC antibody (Monoclonal, Clone RA3- 6B2) | Biolegend, USA | 103229 | IHC (1:500) |
| Antibody | Rat anti-F4/80 antibody (Monoclonal, Clone A3-1) | Biorad, USA | MCA497GA | IHC (1:500) |
| Antibody | Anti-rabbit poly-HRP-conjugated (Polyclonal) | Thermo Fisher, USA | Tyramide Superboost Kit, B40922 | IHC (1X) |
| Antibody | Goat anti-rabbit Alexa Fluor A555 (Polyclonal) | Abcam, UK | Ab150078 | IHC (1:1000) |
| Antibody | Goat anti-rat Alexa Fluor A555 (Polyclonal) | Abcam, UK | Ab150158 | IHC (1:1000) |
| Chemical compound, drug | JWH133 | Tocris, UK | TO-1343 | $CB_2$ receptor agonist |
| Chemical compound, drug | Sodium thiopental | Braun medical, Spain | 635573 | |
| Chemical compound, drug | Isoflurane | Virbac, Spain | 575837–4 | |
| Chemical compound, drug | Paraformaldehyde | Merck Millipore, Germany | 104005 | |
| Chemical compound, drug | DAPI Fluoromount-G mounting media | SouthernBiotech, USA | 0100–20 | |
| Commercial assay or kit | RNeasy Micro kit | Qiagen, Germany | 74004 | |
| Commercial assay or kit | Omniscript reverse transcriptase | Qiagen, Germany | 205111 | |
| Commercial assay or kit | Tyramide Superboost Kit | Thermo Fisher, USA | B40922 | |

*Continued on next page*

*Continued*

| Reagent type (species) or resource | Designation | Source or reference | Identifiers | Additional information |
|---|---|---|---|---|
| Software, algorithm | FACSDiva version 6.2 | BD biosciences, USA | RRID:SCR_001456 | |
| Software, algorithm | Fiji | Wayne Rasband, USA | RRID:SCR_002285 | |
| Software, algorithm | IBM SPSS 19 | IBM Corporation, USA | RRID:SCR_002865 | |
| Software, algorithm | STATISTICA 6.0 | StatSoft, USA | RRID:SCR_014213 | |

## Animals

C57BL/6J male mice were purchased from Charles River Laboratories (L'Arbresle, France), and $CB_2$ knockout male mice defective in the *Cnr2* gene were bred in the Institute of Molecular Psychiatry (University of Bonn, Bonn, Germany). $CB_2$ constitutive knockouts were bred from heterozygous parents and their wild-type littermates were used as controls. Neuron and microglia/macrophage-specific conditional $CB_2$ knockout mice were generated as previously described (*Stempel et al., 2016*). Briefly, mice expressing Cre recombinase under the *Synapsin I* promoter (Syn-Cre+), mice expressing Cre recombinase inserted into the first coding ATG of the *Lyz2* gene (LysM-Cre+) and mice expressing Cre under the promoter of the gene *Scn10a* that codes for Nav1.8 voltage-gated sodium channels (Nav1.8-Cre+) were crossed with *Cnr2* floxed animals ($Cnr2^{fl/fl}$ mice) to obtain Cre+::$Cnr2^{fl/fl}$ mice. These F1 mice (Syn-Cre+-$Cnr2^{fl/-}$, LysM-Cre+-$Cnr2^{fl/-}$ and Nav1.8-Cre+-$Cnr2^{fl/-}$) were backcrossed to $Cnr2^{fl/fl}$ mice to generate mice $Cnr2^{fl/fl}$ and heterozygous for Cre (Cre+-$Cnr2^{fl/fl}$). Syn-Cre+-$Cnr2^{fl/fl}$ (Syn-Cre+), LysM-Cre+-$Cnr2^{fl/fl}$ (LysM-Cre+) and Nav1.8-Cre+-$Cnr2^{fl/fl}$ (Nav1.8-Cre+) mice were selected and further backcrossed to $Cnr2^{fl/fl}$ mice to produce experimental cohorts Syn-Cre+-$Cnr2^{fl/fl}$::$Cnr2^{fl/fl}$, LysM-Cre+-$Cnr2^{fl/fl}$::$Cnr2^{fl/fl}$ and Nav1.8-Cre+-$Cnr2^{fl/fl}$::$Cnr2^{fl/fl}$ containing 50% conditional knockout animals (also referred to as neuronal knockouts or Syn-Cre+, microglial knockouds or LysM-Cre+ and Nav1.8 knockouts or Nav1.8-Cre+) and 50% littermate control animals (referred to as Cre Negative mice throughout the study). Mice defective in the *Trpa1* gene (TRPA1 knockouts, *Kwan et al., 2006*) and their respective control mice (C57BL/6JRccHsd) were bred in the animal facility at Universidad Miguel Hernández (UMH, Elche, Alicante, Spain). For bone-marrow transplantation studies, 2 $CB_2$-GFP BAC mice (*Schmöle et al., 2015a*) or C57BL/6J mice were used as donors and C57BL/6J mice were used as recipient mice. All mice had a C57BL/6J genetic background. The behavioral experimental sequence involving operant self-administration and assessment of nociceptive and anxiety-like behavior was repeated three times in the experiments assessing the effects of JWH133 doses (*Figure 1*) and 4 and 5 times in the experiments evaluating constitutive and conditional knockout mice, respectively (*Figures 2* and *3*). The experiments involving bone-marrow transplantation and lymphocyte depletion were performed once. Sample size was based on previous studies in our laboratory using comparable behavioral approaches (*Bura et al., 2013*; *Bura et al., 2018*; *La Porta et al., 2015*).

The behavioral experiments were conducted in the animal facilities at Universitat Pompeu Fabra (UPF)-Barcelona Biomedical Research Park (PRBB; Barcelona, Spain) and UMH (Elche, Alicante, Spain). Mice were housed in temperature ($21 \pm 1°C$) and humidity-controlled ($55 \pm 10\%$) rooms. For the self-administration experiments, animals were handled during the dark phase of a 12 hr light/dark reverse cycle (light off at 8:00 a.m., light on at 8:00 p.m.). Before starting the experimental procedure, mice were single housed and handled/habituated for 7 days. Food and water were available ad libitum except during the training period for food-maintained operant behavior, when mice were exposed to restricted diet for 8 days. Animal handling and experiments were in accordance with protocols approved by the respective Animal Care and Use Committees of the PRBB, Departament de Territori i Habitatge of Generalitat de Catalunya, UMH and the Institute of Molecular Psychiatry and were performed in accordance with the European Communities Council Directive (2010/63/EU). Whenever possible, animals were randomly assigned to their experimental condition, and experiments were performed under blinded conditions for surgery and pharmacological treatment (*Figure 1*), genotype (*Figures 2* and *3*), bone-marrow transplant and surgery (*Figure 4*), and antibody treatments (*Figure 5*).

## Drugs

JWH133 (Tocris, Bristol, UK) was dissolved in vehicle solution containing 5% dimethyl sulfoxide (Scharlab, Sentmenat, Spain) and 5% cremophor EL (Sigma-Aldrich, Steinheim, Germany) in sterilized water and filtered with a 0.22 µm filter (Millex GP, Millipore, Cork, Ireland). JWH133 was self-administered intravenously (i.v.) at 0.15 or 0.3 mg/kg/infusion in volume of 23.5 µl per injection. In the additional experiments assessing nociceptive behavior in Nav1.8-Cre+ and TRPA1 knockout mice, JWH133 was diluted in a vehicle composed of 5% ethanol (Alcoholes Montplet, Barcelona, Spain), 5% Cremophor EL, and 90% saline (0.9% NaCl; Laboratorios Ern, Barcelona, Spain) to be administered intraperitoneally (i.p.) in a volume of 10 ml/kg. Thiopental (Braun Medical, Barcelona, Spain) was dissolved in saline and administered through the implanted i.v. catheter at 10 mg/kg in a volume of 50 µl.

## Antibody treatment

Anti-ICAM-1 antibody (clone 3E2; 150 µg; BD Biosciences, Franklin Lakes, NJ) and control rabbit IgG (150 µg; Sigma-Aldrich) were dissolved in saline up to a volume of 300 µl as previously reported (*Labuz et al., 2009*), and administered i.p. once a day from the day of the surgery to the last self-administration day.

## Operant self-administration

Mice were first trained for operant food self-administration to facilitate subsequent drug self-administration, as previously described (*Bura et al., 2018*). Briefly, mice were food-restricted for 3 days to reach 90% of their initial weight. Then, mice were trained in skinner boxes (model ENV-307A-CT, Med Associates Inc, Georgia, VT) for 5 days (1 hr session per day) to acquire an operant behavior to obtain food pellets (*Figure 1—figure supplement 1B*, *Figure 2—figure supplement 1B*, *Figure 3—figure supplement 1B*, *Figure 5—figure supplement 1B*). A fixed ratio 1 schedule of reinforcement (FR1) was used, that is 1 nose-poke on the active hole resulted in the delivery of 1 reinforcer together with a light-stimulus for 2 s (associated cue). Nose poking on the inactive hole had no consequence. Each session started with a priming delivery of 1 reinforcer and a timeout period of 10 s right after, where no cues and no reward were provided following active nose-pokes. Food sessions lasted 1 hr or until mice nose-poked 100 times on the active hole, whichever happened first. After the food training, mice underwent a partial sciatic nerve ligation (PSNL) or a sham surgery, and 4 days later an i.v. catheter was implanted in the right jugular vein to allow drug delivery. Mice started the drug self-administration sessions 7 days after the PSNL/sham surgery. In these sessions, the food reinforcer was substituted by drug/vehicle infusions. Self-administration sessions were conducted during 12 consecutive days, and mice received JWH133 (0.15 or 0.3 mg/kg) or vehicle under FR1 (*Figure 1—figure supplement 1B*, *Figure 2—figure supplement 1B*, *Figure 3—figure supplement 1B*, *Figure 5—figure supplement 1B*). Sessions lasted 1 hr or until 60 active nose-pokes. Active and inactive nose-pokes were recorded after each session and discrimination indices were calculated as the difference between the nose pokes on the active and the inactive holes, divided by the total nose pokes. Data from the last three drug self-administration sessions was used for statistical analysis to exclude interference with food-driven operant behavior.

## Partial Sciatic Nerve Ligation

Mice underwent a partial ligation of the sciatic nerve at mid-thigh level to induce neuropathic pain, as previously described (*Malmberg and Basbaum, 1998*) with minor modifications. Briefly, mice were anaesthetized with isoflurane (induction, 5% V/V; surgery, 2% V/V) in oxygen and the sciatic nerve was exposed at the level of the mid-thigh of the right hind leg. At ~1 cm proximally to the nerve trifurcation, a tight ligature was created around 33–50% of the cranial side of the sciatic nerve using a 9–0 non-absorbable virgin silk suture (Alcon Cusí SA, Barcelona, Spain) and leaving the rest of the nerve untouched. The muscle was then stitched with 6–0 silk (Alcon Cusí), and the incision was closed with wound clips. Sham-operated mice underwent the same surgical procedure except that the sciatic nerve was not ligated.

## Catheterization

Mice were implanted with indwelling i.v. silastic catheter, as previously reported (*Soria et al., 2005*). Briefly, a 5.5 cm length of silastic tubing (0.3 mm inner diameter, 0.64 mm outer diameter; Silastic, Dow Corning Europe, Seneffe, Belgium) was fitted to a 22-gauge steel cannula (Semat Technical Ltd., Herts, UK) that was bent at a right angle and then embedded in a cement disk (Dentalon Plus, Heraeus Kulzer, Wehrheim, Germany) with an underlying nylon mesh. The catheter tubing was inserted 1.3 cm into the right jugular vein and anchored with suture. The remaining tubing ran sub-cutaneously to the cannula, which exited at the midscapular region. All incisions were sutured and coated with antibiotic ointment (Bactroban, GlaxoSmithKline, Madrid, Spain).

## Nociception

Sensitivity to heat and mechanical stimuli were used as nociceptive measures of neuropathic pain. Ipsilateral and contralateral hind paw withdrawal thresholds were evaluated the day before, 3 and 6 days after the nerve injury, as well as after the last self-medication session on day 18. Heat sensitivity was assessed by recording the hind paw withdrawal latency in response to radiant heat applied with the plantar test apparatus (Ugo Basile, Varese, Italy) as previously reported (*Hargreaves et al., 1988*). Punctate mechanical sensitivity was quantified by measuring the withdrawal response to von Frey filament stimulation through the up–down paradigm, as previously reported (*Chaplan et al., 1994*). Filaments equivalent to 0.04, 0.07, 0.16, 0.4, 0.6, 1 and 2 g were used, applying first the 0.4 g filament and increasing or decreasing the strength according to the response. The filaments were bent and held for 4–5 s against the plantar surface of the hind paws. Clear paw withdrawal, shaking or licking was considered a nociceptive-like response. Four additional filaments were applied since the first change of response (from negative to positive or from positive to negative), once each time. The sequence of the last six responses was used to calculate the withdrawal threshold following the method described by *Dixon, 1965*.

## Anxiety-like behavior

Anxiety-like behavior was evaluated with an elevated plus maze made of Plexiglas and consisting of four arms (29 cm long x 5 cm wide), two open and two closed, set in cross from a neutral central square (5 × 5 cm) elevated 40 cm above the floor. Light intensity in the open and closed arms was 45 and 5 lux, respectively. Mice were placed in the neutral central square facing 1 of the open arms and tested for 5 min. The percentage of entries and time spent in the open and closed arms was determined.

## RNA extraction and reverse transcription

Ipsilateral L3-L4 dorsal root ganglia from mice of the ICAM-1 experiment were collected on day 20 after the PSNL. Samples were rapidly frozen in dry ice and stored at −80°C. Isolation of total RNA was performed using the RNeasy Micro kit (Qiagen, Stokach, Germany) according to the manufacturer's instructions. Total RNA concentration was measured using a NanoDrop ND-1000 Spectrophotometer (NanoDrop Technologies Inc, Montchanin, DE). RNA quality was determined by chip-based capillary electrophoresis using an Agilent Bioanalyzer 2100 (Agilent, Palo Alto, CA). Reverse transcription was performed using Omniscript reverse transcriptase (Qiagen) at 37°C for 60 min.

## Quantitative real-time PCR analysis

The qRT-PCR reactions were performed using Assay-On-Demand TaqMan probes: Hprt1 Mm01545399_m1, CD2 Mm00488928 m1, CD4 Mm00442754_m1, CD19 Mm00515420_m1, C1q Mm00432162_m1 (Applied Biosystems, Carlsbad, CA) and were run on the CFX96 Touch Real-Time PCR machine (BioRad, Hercules, CA). Each template was generated from individual animals, and amplification efficiency for each assay was determined by running a standard dilution curve. The expression of the Hprt1 transcript was quantified at a stable level between the experimental groups to control for variations in cDNA amounts. The cycle threshold values were calculated automatically by the CFX MANAGER v.2.1 software with default parameters. RNA abundance was calculated as $2^{-(Ct)}$. Levels of the target genes were normalized against the housekeeping gene, Hprt1, and compared using the ΔΔCt method (*Livak and Schmittgen, 2001*).

## Bone marrow transplantation

C57BL/6J mice received bone marrow from $CB_2$-GFP BAC or C57BL/6J male mice. G-irradiation of C57BL/6J recipient male mice (9.5 Gy) was performed in a 137Cs-g IBL 437 C H irradiator (Schering CIS Bio international) at 2.56 Gy/min rate in order to suppress their immune response. Afterwards, approximately $5 \times 10^5$ bone marrow cells collected from donors ($CB_2$-GFP BAC or C57BL/6J) and transplanted through the retro-orbital venous sinus of the recipient mice. Irradiated mice were inspected daily and were given 150 ml of water with enrofloxacin at 570 mg/l and pH 7.4 (Bayer, Germany) for 30 days to reduce the probability of infection from opportunistic pathogens. Peripheral blood samples (150 µl) were collected by tail bleeding into a tube with 0.5 M EDTA solution to evaluate immune system recovery through flow cytometry 4, 8 and 12 weeks after the bone marrow transplantation.

## Flow cytometry

For the analyses of hematopoietic cells, a hypotonic lysis was performed to remove erythrocytes. 50 µl of blood was lysed using 500 µl of ACK (Ammonium-Chloride-Potassium) Lysing Buffer (Lonza, Walkersville) 10 min at room temperature. After the erythrocytes lysis, two washes with PBS were performed prior the incubation with the antibodies for 30 min at 4°C. Cells were stained with the following fluorochrome-coupled antibodies: Allophycocyanin (APC)-conjugated anti-mouse CD11b (1:300; cn.17–0112 eBioscience, USA) to label myeloid cells, phycoerythrin (PE)-conjugated anti-mouse B220 (1:100; cn.12–0452, eBioscience, USA) for B lymphocytes and phycoerythrin/cyanine (PE/Cy7)-conjugated anti-mouse CD3, 1:100; cn.100320, BioLegend, USA) for T lymphocytes. Immunofluorescence of labeled cells was measured using a BD LSR II flow cytometer. Dead cells and debris were excluded by measurements of forward- versus side-scattered light and DAPI (4′,6-diamino-2-phenylindole) (Sigma) staining. Gates for the respective antibodies used were established with isotype controls and positive cell subset controls. Data analysis was carried out using FACSDiva version 6.2 software (BD biosciences).

## Immunohistochemistry

Mice were sacrificed 2 weeks after the PSNL/sham surgery and L3-L5 dorsal root ganglia were collected to quantify GFP+ cells in mice transplanted with bone marrow cells of $CB_2$-GFP or C57BL6/J mice. Ganglia were freshly extracted and fixed in 4% paraformaldehyde during 25 min at 4°C. After $3 \times 5$ min washes with phosphate buffered saline (PBS) 0.1 M (pH 7.4), were preserved overnight in a 30% sucrose solution in PBS 0.1 M containing sodium azide 0.02%. 24 hr later, ganglia were embedded in molds filled with optimal cutting temperature compound (Sakura Finetek Europe B.V., Netherlands) and frozen at −80°C. Samples were sectioned with a cryostat at 10 µm, thaw-mounted on gelatinized slides and stored at −20°C until use. Dorsal root ganglia sections were treated 1 hr with 0.3 M glycine, 1 hr with oxygenated water 3% (Tyramide Superboost Kit, B40922, Thermo Fisher, USA) and, after $3 \times 5$ min washes with PBS 0.01 M, 1 hr with blocking buffer. Samples were incubated 16 hr at room temperature with rabbit anti-GFP (1:2000, A11122, Thermo Fisher, USA) antibody. After $3 \times 10$ min washes with PBS 0.01 M, sections were incubated with anti-rabbit poly-HRP-conjugated secondary antibody for 1 hr and washed $4 \times 10$ min. Alexa Fluor tyramide reagent was applied for 10 min and then the Stop Reagent solution for 5 min (Tyramide Superboost Kit). Afterwards samples were incubated 2 hr at room temperature with primary antibodies diluted in blocking buffer (PBS 0.01 M, Triton X-100 0.3%, Normal Goat Serum 10%). The following primary antibodies were used: rabbit anti-peripherin (1:200, PA3-16723, Thermo Fisher, USA), rabbit anti-β-III tubulin (1:1000, ab18207, Abcam, UK), rat anti-CD45R/B220 APC (1:500, Clone RA3-6B2, 103229, Biolegend, USA) and rat anti-F4/80 (1:500, Clone A3-1, MCA497GA, Biorad, USA). After $3 \times 5$ min washes, all sections were treated with goat secondary antibodies from Abcam (UK) for 1 hr at room temperature: anti-rabbit Alexa Fluor 555 (1:1000, ab150078) and anti-rat Alexa Fluor 555 (1:1000, ab150158). Samples were then washed with PBS 0.01 M and mounted with $24 \times 24$ mm coverslips (Brand, Germany) using Fluoromount-G with DAPI (SouthernBiotech, USA).

## Microscope image acquisition and processing

Confocal images were taken with a Leica TCS SP5 confocal microscope (Leica Microsystems, Mannheim, Germany) on a DM6000 stand using $20 \times 0.7$ NA Air and $63 \times 1.4$ NA Oil Immersion Plan

Apochromatic lenses. Leica Application Suite Advanced Fluorescence software (Leica Microsystems, Mannheim, Germany) was used to acquire the images and DAPI, Alexa 488 and Alexa 555 channels were taken sequentially. Images of DAPI were taken with 405 nm excitation and emission detection between 415 and 480 nm; images of Alexa 488 were taken with 488 nm excitation and emission detection between 495 and 540 nm; and images of Alexa 555 were taken with 543 nm excitation and emission detection between 555 and 710 nm. Room temperature was kept at $22 \pm 1°C$ during all imaging sessions. All images were equally processed and quantified with Fiji software (National Institutes of Health, USA). To determine the percentage of dorsal root ganglia area occupied by GFP (+) neurons auto-threshold ('Otsu') was set between 0–50 in all images and then converted to mask. Afterwards, operations included Close, Fill holes and Watershed neurons for separation and particles between 100–100000 pixel units and circularity 0.2–1.0 were counted. To analyze the number of GFP+ cells per dorsal root ganglia area, background was subtracted from all images (rolling = 5), set to an auto-threshold ('Default') between 0–70 and converted to mask. Particles considered GFP+ cells were nucleated, 7–100 microns$^2$ and 0.9–1.0 circularity.

## Statistical analysis

Self-administration and nociceptive behavioral data were analyzed using a linear mixed model with three (surgery, day and dose) or two factors (day and genotype or antibody treatment) and their interactions. For the covariance structure of the repeated measures, a diagonal matrix was chosen. Bonferroni post hoc analysis was performed when pertinent. Areas Under the Curve (AUCs) of time-courses for operant responding were analyzed using two-way analysis of variance (ANOVA). Active and inactive responses were analyzed taking into account surgery and dose effects in the dose-response experiments, and active/inactive and genotype or antibody treatment in the knockout and antibody experiments. Anxiety-like behavior was analyzed using two-way ANOVA (surgery and dose for dose-response experiments), one-way ANOVA (genotype of conditional knockouts) or t-tests (constitutive knockout and antibody treatment), followed by Bonferroni adjustments when required. Mechanical and thermal thresholds in Nav1.8 and TRPA1 knockout mice treated with JWH133 were analyzed using three-way repeated measures ANOVA with surgery and genotype as between-subject factors and treatment as within-subject factor, followed by Bonferroni post-hoc test when appropriate. Immunohistochemistry of bone marrow-transplanted mice was analyzed using the Bonferroni-Dunn's test to adjust for multiple comparisons after multiple t-tests, and qPCR results after antibody treatments were compared with t-tests. IBM SPSS 19 (SPSS Inc, Chicago, IL) and STATISTICA 6.0 (StatSoft, USA) software were used to analyze the data, and differences were considered statistically significant when p value was below 0.05. All experimental data and statistical analyses of this study are included in the manuscript and its supplementary files. Raw data and results of statistical analyses are provided in the respective Source Data Files and their containing data sheets.

## Acknowledgements

Financial support of European Commission [NeuroPain, FP7-602891-2], Instituto de Salud Carlos III, Redes temáticas de investigación cooperativa en salud – Red de trastornos adictivos [#RD16/0017/0020/FEDER], 'Ministerio de Ciencia, Innovación y Universidades' [#SAF2017-84060-R FEDER] and 'AGAUR' [SGR2017-669, Institució Catalana de Recerca i Estudis Avançats Academia Award 2015] to RM, 'Generalitat de Catalunya- Agència de Gestió d'Ajuts Universitaris i de Recerca-AGAUR' [#2018 FI_B 00207], Polish Ministry of Science and Education [#3070/7.PR/2014/2], RTI2018-097189-B-C21 and UMH-PAR2019. Authors thank Itzel M Lara, Hugo Ramos, Roberto Cabrera and Cristina Fernández for their help and technical expertise.

## Additional information

### Funding

| Funder | Grant reference number | Author |
|---|---|---|
| European Commission | NeuroPain, FP7-602891-2 | Rafael Maldonado |

| Instituto de Salud Carlos III | RTA, RD16/0017/0020/FEDER | Rafael Maldonado |
| Ministerio de Ciencia, Innovación y Universidades | SAF2017-84060-R FEDER | Rafael Maldonado |
| Generalitat de Catalunya | SGR2017-669, ICREA Academia Award 2015 | Rafael Maldonado |
| Generalitat de Catalunya | 2018 FI_B 00207 | Angela Ramírez-López |
| Polish Ministry of Science and Education | 3070/7.PR/2014/2 | Ryszard Przewlocki |
| Spanish Ministry of Science, Innovation and Universities | 2018-097189-B-C21 | Antonio Ferrer-Montiel |
| Universidad Miguel Hernandez | UMH-PAR2019 | Antonio Ferrer-Montiel |

The funders had no role in study design, data collection and interpretation, or the decision to submit the work for publication.

### Author contributions
David Cabañero, Conceptualization, Data curation, Formal analysis, Supervision, Validation, Investigation, Visualization, Methodology, Writing - original draft, Writing - review and editing; Angela Ramírez-López, Data curation, Formal analysis, Supervision, Validation, Investigation, Visualization, Methodology, Writing - review and editing; Eva Drews, Anne Schmöle, David M Otte, Resources, Validation, Methodology; Agnieszka Wawrzczak-Bargiela, Validation, Investigation, Visualization, Methodology; Hector Huerga Encabo, Validation, Investigation, Methodology; Sami Kummer, Validation, Methodology; Antonio Ferrer-Montiel, Resources, Writing - review and editing; Ryszard Przewlocki, Resources, Formal analysis, Supervision, Funding acquisition, Investigation, Visualization, Methodology, Writing - review and editing; Andreas Zimmer, Resources, Supervision, Validation, Investigation, Methodology, Writing - review and editing; Rafael Maldonado, Conceptualization, Resources, Data curation, Supervision, Funding acquisition, Validation, Investigation, Visualization, Methodology, Project administration, Writing - review and editing

### Author ORCIDs
David Cabañero (iD) https://orcid.org/0000-0002-1133-0908
Rafael Maldonado (iD) https://orcid.org/0000-0002-4359-8773

### Ethics
Animal experimentation: Animal handling and experiments were in accordance with protocols approved by the respective Animal Care and Use Committees of the PRBB, Departament de Territori i Habitatge of Generalitat de Catalunya and the Institute of Molecular Psychiatry and were performed in accordance with the European Communities Council Directive (2010/63/EU).

### Decision letter and Author response
Decision letter https://doi.org/10.7554/eLife.55582.sa1
Author response https://doi.org/10.7554/eLife.55582.sa2

# Additional files

### Supplementary files
• Transparent reporting form

### Data availability
All experimental data and statistical analyses of this study are included in the manuscript and its supplementary files. Raw data and results of statistical analyses are provided in the Source Data File and its containing data sheets.

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
