## [Decision Letter]

**Acceptance summary:**

To what extent cannabinoids are effective against chronic pain is still unclear. Using self-administration of a selective CB_2_ receptor antagonist in mice, the authors report that expression of the CB_2_ receptor in both neurons and lymphocytes underlies a complex interaction that can regulate spontaneous pain following nerve injury.

**Decision letter after peer review:**

Thank you for submitting your article "Protective role of neuronal and lymphoid cannabinoid CB_2_ receptors in neuropathic pain" for consideration by *eLife*. Your article has been reviewed by two peer reviewers, and the evaluation has been overseen by a Reviewing Editor and Christian Büchel as the Senior Editor. The reviewers have opted to remain anonymous.

The reviewers have discussed the reviews with one another and the Reviewing Editor has drafted this decision to help you prepare a revised submission.

Both reviewers found many positive aspects to your manuscript. However, although reviewer #1 believes that the concerns could be addressed without providing additional experimental data, reviewer 2 has identified some significant concerns that do need additional information. The specific experiments that are indicated address the underlying mechanism, including the extent to which receptor expression on sensory neurons is involved and most importantly, the contribution of TRPA1. We appreciate that the present COVID-19 pandemic will make it impossible to complete the requested studies within the normal 2 month period, which will involve new mouse crosses, we are willing to accept a revised manuscript when you are able to return to the laboratory and complete the studies.

Reviewer #1:

This is a very interesting paper. While demonstration of CB_2_ receptor agonist self-administration in rodent models of chronic pain is not in itself novel, there is a sufficient body of additional novel and exciting work in this paper to set it apart from previously published work. In particular, the mechanistic dissection using tissue-specific KO mice, coupled with the demonstration of that CB_2_ receptor-expressing lymphocytes infiltrate peripheral neurons, and to a greater degree in nerve-injured versus sham mice. The anxiety-related results are also very interesting. The paper is well-written and the results, for the most part, are clear.

1) While there are significant novel results and important conclusions that can be and have been drawn from the work, the mechanism underlying the increased self-administration of JWH133 in PSNL mice has still not been fully elucidated. The authors have shown it is CB_2_ receptor-dependent, but not due to CB_2_ receptors in neurons or monocyte-derived cells. Neither does it appear to be due to CB_2_ receptors on infiltrating lymphocytes. So the question still remains as to what mechanism or target is mediating the effect. I think the Discussion should address this limitation in more detail, and put forward some potential mechanisms.

2) Is it possible that there could be some involvement for CB_1_ receptors? JWH133 is relatively selective for CB_2_ over CB_1_, but to my knowledge it does still have some affinity for (and potential activity at) CB_1_. How can the authors rule out a potential involvement of CB_1_ in the self-administration of JWH133 after PSNL.

3) Given the anxiety-related aspect (and indeed the self-administration/pain aspect), why did the authors not look at whether lymphocytes expressing CB_2_ also infiltrate brain neurons? It would be very interesting to know if they infiltrate neurons in brain regions such as the amygdala, PFC, PAG and other regions known to be important in pain, anxiety and drug self-administration in pain models.

Reviewer #2:

This is an interesting paper that addresses and important and timely subject, namely identification of novel non opioid approaches to pain management. The authors direct their attention the CB_2_ receptor. Using perhaps the best characterized, and quite selective CB_2_ ligand, the authors implicated CB_2_ receptors in both neuronal and non-neuronal cells in the spontaneous pain that occurs in a partial nerve injury model in the mouse. The studies largely used either mice in which the CB_2_ receptor was deleted in all cells only in neurons, or selectively in monocyte derived cells.

There are many intriguing findings in the paper, however, one is left with the feeling that there are hints at mechanisms, but nothing definitive is established. And major questions, which I believe could have been addressed with more selective Cre-mediated deletion of the receptor, are never answered. Hints here and there, but nothing definitive.

The authors focus on sensory neurons and the non-neuronal cells that surround the neuronal cell bodies in the DRG. How might the neurons in the DRG, which they appear to presume must be mediating the input that drives spontaneous pain, not be relevant to acute pain processing? The previous report of Soethoudt et al., 2017, which defined the specificity of JWH133, found that this compound is without effect on acute pain even at doses up to 100mg/kg. I am not sure how to translate that dose to the iv administration in the present paper, but my assumption is that the 100mg/kg dose is at least equivalent. Granted JWH133 is not potent, but then how does it affect spontaneous pain?

The most straightforward test of the DRG neurons is to delete the CB_2_ receptor from neurons, using one of several selective Cre lines (e.g. NaV1.8-Cre). This is particularly important as the authors highlight the apparent translocation of the CB_2_ receptor from non-neuronal cells around the DRG to neurons. But they only used peripherin to mark the neurons, so that any change in myelinated afferents would be missed. As neurons are the only structure that can get information into the spinal cord, do the authors propose that it is this small 4% of DRG neurons that is key?

Perhaps the most glaring piece missing as to mechanism is the fact, acknowledged by the authors, that JWH133 has a significant action at TRPA1, which is expressed by sensory neurons. Most importantly, the authors found that the CB2R null only had 50% reduced nose poke for JWH133. Clearly, JWH133 must exert its effect, at least in part, on another target. In their Abstract, the authors are very careful concerning this finding, writing that ""While constitutive deletion of CB2r disrupted JWH133-taking behavior….". In other words JWH133 disrupted, it did not prevent or eliminate the behavior. So clearly, there is something else mediating JWH133's effects. Studying JWH133 effects in the TRPA1 mutant, and ideally in the mouse in which TRPA1 is selectively deleted from sensory neurons, is critical to understanding this drug's actions. Results from that study would add greatly to the authors study.

Also, as TRPA1 is expressed in sensory neurons, are they now proposing that TRPA1 only contributes to spontaneous pain? That is certainly not the case. In fact, a previous study did report that JWH133 blocked pain behaviors provoked by AITC, a TRPA1 agonist. A simple experiment would be to test the effect of JWH133 against 0.5% formalin evoked nocifensive behaviors. As for AITC, 0.5% formalin evoked behaviors are lost in the TRPA1 ko.

Another simple experiments that would get at mechanism is to examine the effect of JWH133 on nerve injury provoked microglial activation in the dorsal horn. If a decreased activation were demonstrated, the case would be much stronger that the drug is acting on sensory neurons.

Concerns about the immunohistochemistry: First, the images presented are not all that convincing, and particularly difficult to read given that the percentage of double labeled neurons is small. It is also very odd that the authors had to use TSA amplification. Also there is no mention of controls for antibody specificity.

---

## [Author Response]

Reviewer #1:[…] 1) While there are significant novel results and important conclusions that can be and have been drawn from the work, the mechanism underlying the increased self-administration of JWH133 in PSNL mice has still not been fully elucidated. The authors have shown it is CB_2_ receptor-dependent, but not due to CB_2_ receptors in neurons or monocyte-derived cells. Neither does it appear to be due to CB_2_ receptors on infiltrating lymphocytes. So the question still remains as to what mechanism or target is mediating the effect. I think the Discussion should address this limitation in more detail, and put forward some potential mechanisms.

We agree with the reviewer that this is a limitation of the work and the following paragraph was added to the Discussion section, as requested:

“While we identify CB_2_-expressing neurons and lymphocytes as cellular entities involved in spontaneous and evoked neuropathic pain, the efficacy of the CB_2_ agonist eliciting its own self-administration to alleviate pain was only disrupted in constitutive CB_2_ knockout mice. […] Vascular cells may represent alternative participants of this behavior since JWH133 showed local vasodilatory effects (McDougall and Thomson, 2008) and endothelial functional CB_2_ receptor was found in cerebral microvasculature (Ramirez et al., 2012; Onaivi et al., 2012).”

We also explained further the self-administration results on the neuronal knockouts to highlight neuronal involvement on affective-motivational responses:

“Thus, increased self-administration suggests an enhanced affective-motivational component of pain and not reduced drug efficacy on this aspect, since nerve-injured C57BL6/J mice exposed to the low JWH133 dose did not show compensatory increased self–administration (Figure 1).”

2) Is it possible that there could be some involvement for CB_1_ receptors? JWH133 is relatively selective for CB_2_ over CB_1_, but to my knowledge it does still have some affinity for (and potential activity at) CB_1_. How can the authors rule out a potential involvement of CB_1_ in the self-administration of JWH133 after PSNL.

Partial involvement of CB_1_ receptor activity in JWH133 self-administration after PSNL cannot be completely ruled out from our data. We included this paragraph in the Discussion section to take into account this possibility:

“Another possibility is a minor involvement of CB_1_ receptor in JWH133 self-administration after the nerve injury, since this compound is a selective CB_2_ agonist that exhibited 40-fold higher affinity for mouse CB_2_ than for CB_1_ receptor (Soethoudt et al., 2017). […] Since JWH133 selectivity for human CB_2_ vs. human CB_1_ is higher (153-fold selectivity), we do not foresee any concern with CB_1_-related behavioral effects using similar molecules in humans.”

3) Given the anxiety-related aspect (and indeed the self-administration/pain aspect), why did the authors not look at whether lymphocytes expressing CB_2_ also infiltrate brain neurons? It would be very interesting to know if they infiltrate neurons in brain regions such as the amygdala, PFC, PAG and other regions known to be important in pain, anxiety and drug self-administration in pain models.

We agree on the interest of determining lymphocyte infiltration in pain, anxiety and reinforcement-related brain regions and the following sentence was included in the Discussion section:

“Our data also underscore the interest of investigating the role of lymphoid cells in brain regions involved in pain, anxiety or negative reinforcement during chronic neuroinflammatory processes.”

Given the previous results demonstrating hippocampal neuronal CB_2_ receptor activity (Stempel et al., 2016), we looked for mRNA changes of neuronal and immune markers in the hippocampus of the CB_2_ knockout mice and their control littermates. However, the negative results obtained in our experimental conditions (see Author response image 1) discouraged us of continuing biochemical determinations in this region.

**Author response image 1. sa2fig1:** 

Reviewer #2:[…] There are many intriguing findings in the paper, however, one is left with the feeling that there are hints at mechanisms, but nothing definitive is established. And major questions, which I believe could have been addressed with more selective Cre-mediated deletion of the receptor, are never answered. Hints here and there, but nothing definitive.The authors focus on sensory neurons and the non-neuronal cells that surround the neuronal cell bodies in the DRG. How might the neurons in the DRG, which they appear to presume must be mediating the input that drives spontaneous pain, not be relevant to acute pain processing? The previous report of Soethoudt et al., 2017, which defined the specificity of JWH133, found that this compound is without effect on acute pain even at doses up to 100mg/kg. I am not sure how to translate that dose to the iv administration in the present paper, but my assumption is that the 100mg/kg dose is at least equivalent. Granted JWH133 is not potent, but then how does it affect spontaneous pain?

DRG neurons could certainly mediate spontaneous pain after peripheral nerve injury in our experimental conditions. However, peripheral CB_2_ do not play an important role in acute pain processing (Hanus et al., 1999; Naguib et al., 2008). Indeed, CB_2_ agonists have poor efficacy in the absence of ongoing inflammatory conditions (Bie et al., 2018), which emphasizes the relevance of the presence of pathological conditions for the effects of JWH133. Since the mouse cannabinoid triad in the Soethoudt paper was in principle conducted in naïve mice exposed to JWH133, a general lack of efficacy could be expected even considering the high doses used. In our experiments in nerve-injured mice, 5 and 10 mg/kg i.p. produced significant mechanical antinociception above 50% MPE, and we previously observed significant effects with 1 and 5 mg/kg i.p. during chronic knee inflammation (La Porta et al., Pain, 2015). Both inflammatory and neuropathic pain conditions are different to those used by Soethoudt et al., 2017, to evaluate acute pain in naïve mice. According to the Soethoudt paper, i.v. doses of 1.6 mg/kg have a half-life of 1.1 hour. The i.v. dose used in our work is lower (0.3 mg/kg) and should induce short half-life sub-maximal responses sufficient to promote contingent and repeated operant self-administration in order to relieve non-evoked pain under these neuropathic pain conditions.

The most straightforward test of the DRG neurons is to delete the CB_2_ receptor from neurons, using one of several selective Cre lines (e.g. NaV1.8-Cre). This is particularly important as the authors highlight the apparent translocation of the CB_2_ receptor from non-neuronal cells around the DRG to neurons.

We conducted an additional experiment to evaluate JWH133 antinociceptive effects in CB_2_ Nav1.8-Cre+ mice lacking CB_2_ receptor in Nav1.8 peripheral neurons. Conditional knockout mice and their wild-type littermates were subjected to the partial sciatic nerve ligation or the sham procedure and tested for mechanical and thermal hypersensitivity after administration of vehicle or JWH133 at doses of 5 and 10 mg/kg. We observed similar effects of JWH133 inhibiting mechanical and thermal hypersensitivity of Nav1.8-Cre+ and Cre Negative mice. Together with the results obtained in the Syn-Cre+ mice and the bone-marrow transplantation experiments, these results suggest the involvement of CB_2_ receptor from central locations and/or of CB_2_ receptor translocated from non-neuronal cells. The results are included in Figure 3—figure supplement 2.

The line of mice used to genetically disrupt CB_2_ receptors in Nav1.8 cells was less sensitive than other mouse lines to promote thermal hyperalgesia after sciatic nerve injury and was therefore less sensitive to the anti-hyperalgesic effect of JWH133 in the plantar test under our experimental conditions, although a significant general effect of JWH133 was revealed at the dose of 10 mg/kg in both genotypes. Average baseline thresholds or contralateral paw thresholds did not show significant differences between both strains of mice subjected to the neuropathic pain model (see Author response image 2).

A new figure supplement was added to Figure 3 (Figure 3—figure supplement 2). Details of the methods were included (Materials and methods subsections “Animals”, “Drugs” “Statistical analysis”), results were added to the Results section (subsection “Participation of neuronal and monocyte CB_2_ receptor in neuropathic pain symptomatology”) and source data files (Figure 3—figure supplement 2—source data 1), and discussed in the Discussion section (third paragraph).

But they only used peripherin to mark the neurons, so that any change in myelinated afferents would be missed. As neurons are the only structure that can get information into the spinal cord, do the authors propose that it is this small 4% of DRG neurons that is key?

As indicated in the Materials and methods section (subsection “Immunohistochemistry”), we used a mixture of anti-peripherin and anti-β-III tubulin antibodies: “rabbit anti-peripherin (1:200, PA3-16723, Thermo Fisher, USA), rabbit anti-β-III tubulin (1:1000, ab18207, Abcam, UK)”. In the previous version of the manuscript, we included by mistake in the Materials and methods the anti-neurofilament heavy antibody and this mistake has been now corrected. The rationale of using both anti-peripherin and anti-β-III tubulin was to mark all myelinated and non-myelinated neurons, since this anti β-III tubulin is known to miss some percentage of unmyelinated neurons and peripherin only labels unmyelinated ones. We apologize for this mistake in the Materials and methods section of the previous version of the manuscript.

Perhaps the most glaring piece missing as to mechanism is the fact, acknowledged by the authors, that JWH133 has a significant action at TRPA1, which is expressed by sensory neurons. Most importantly, the authors found that the CB2R null only had 50% reduced nose poke for JWH133. Clearly, JWH133 must exert its effect, at least in part, on another target. In their Abstract, the authors are very careful concerning this finding, writing that "While constitutive deletion of CB2r disrupted JWH133-taking behavior….". In other words JWH133 disrupted, it did not prevent or eliminate the behavior. So clearly, there is something else mediating JWH133's effects. Studying JWH133 effects in the TRPA1 mutant, and ideally in the mouse in which TRPA1 is selectively deleted from sensory neurons, is critical to understanding this drug's actions. Results from that study would add greatly to the authors study.

An additional experiment was performed to compare JWH133 efficacy in wild-type (WT) and TRPA1 constitutive knockout mice (TRPA1 KO) under a C57BL/6J background. Mice were subjected to the partial sciatic nerve ligation or to the sham procedure and after 7 days were tested for mechanical and thermal hypersensitivity to evaluate the effect of vehicle and JWH133 at the doses of 5 and 10 mg/kg. We observed similar effects of JWH133 inhibiting mechanical hypersensitivity in TRPA1 KO and WT mice. Interestingly, TRPA1 KO mice showed a prominent inhibition of neuropathic thermal hypersensitivity. In spite of this lack of sensitivity, a significant general JWH133 effect was observed in nerve-injured mice also in thermal sensitivity, regardless of the genotype of the mice.

The results on mechanical sensitivity show that the effects of the CB_2_ receptor agonist are retained in the TRPA1 KO mice, suggesting that these effects are not due to an interaction of the drug with the TRPA1 receptor. The lack of thermal hypersensitivity observed in the TRPA1 KO mice may occlude possible JWH133 effects on neuropathic thermal hyperalgesia through TRPA1. However, a significant effect of JWH133 was observed in both strains of mice after the nerve injury (10 mg/kg dose), showing that at least the CB_2_ receptor is involved in the inhibitory effect on thermal hyperalgesia. Baseline or contralateral paw thresholds did not show significant differences between TRPA1 KO and WT mice subjected to the neuropathic pain model (Author response image 3).

**Author response image 3. sa2fig3:** 

A new figure supplement was added to Figure 2 (Figure 2—figure supplement 2). Details of the methods were incorporated (Materials and methods subsections “Animals”, “Drugs” and “Statistical analysis”) and results were added to the Results section (subsection “CB_2_ receptor mediates JWH133 effects on spontaneous pain alleviation”) and source data file (Figure 2—figure supplement 2—source data 1), and discussed in the Discussion section (second paragraph).

Also, as TRPA1 is expressed in sensory neurons, are they now proposing that TRPA1 only contributes to spontaneous pain? That is certainly not the case. In fact, a previous study did report that JWH133 blocked pain behaviors provoked by AITC, a TRPA1 agonist. A simple experiment would be to test the effect of JWH133 against 0.5% formalin evoked nocifensive behaviors. As for AITC, 0.5% formalin evoked behaviors are lost in the TRPA1 ko.

These studies and our additional new results using TRPA1 knockout mice were added to the Discussion section:

“Our results with the nerve-injured TRPA1 knockout mice revealed that JWH133 preserves its efficacy inducing antinociception in the absence of this receptor. […] Interestingly, TRPA1 deletion prevented the development of thermal hyperalgesia after the nerve injury in our study and formalin (0.5%) evoked nocifensive behaviors are also lost in the TRPA1 knockout (McNamara et al., 2007).”

Another simple experiments that would get at mechanism is to examine the effect of JWH133 on nerve injury provoked microglial activation in the dorsal horn. If a decreased activation were demonstrated, the case would be much stronger that the drug is acting on sensory neurons.

Previous studies have reported a relevant role of CB_2_ receptor activity in the control of nociceptive responses through central areas such as the rostral ventromedial medulla (Li et al., 2017; Song et al., 2018). Therefore, an effect of JWH133 on spinal microglial activation could be the result of modified descending input from this pain-modulating region. These previous results also underline a broad role of microglia in pain control that seems to be not only limited to the spinal dorsal horn.

Concerns about the immunohistochemistry: First, the images presented are not all that convincing, and particularly difficult to read given that the percentage of double labeled neurons is small. It is also very odd that the authors had to use TSA amplification. Also there is no mention of controls for antibody specificity.

Additional images showing higher levels of double-labeled neurons have been added. They constitute now Figure 4—figure supplement 5 (A, CB_2_-GFP Sham; B, CB_2_-GFP PSNL) and Figure 4—figure supplement 6 (Amplified Merged Images of CB_2_-GFP Sham and CB_2_-GFP PSNL from Figure 4—figure supplement 5).

We used Tyramide Signal Amplification (TSA) due to low signal-to-background ratio, which could be associated to the sample processing or to low expression levels. Specificity tests for TSA are shown in Figure 4—figure supplement 7. Controls for antibody specificity are included as Figure 4—figure supplement 8. All supplements are mentioned in the legend of Figure 4.

References:

Hanus L, Breuer A, Tchilibon S, Shiloah S, Goldenberg D, Horowitz M, Pertwee RG, Ross RA, Mechoulam R, Fride E. 1999. HU-308: A specific agonist for CB_2_, a peripheral cannabinoid receptor. Proc Natl Acad Sci U.S.A. 96:14228-14223. Doi: 10.1073/pnas.96.25.14228

Li MH, Suchland KL, Ingram SL. 2017. Compensatory Activation of Cannabinoid CB_2_ Receptor Inhibition of GABA Release in the Rostral Ventromedial Medulla in Inflammatory Pain. J Neurosci 37:626-636. doi:10.1523/JNEUROSCI.1310-16.2016

McNamara CR, Mandel-Brehm J, Bautista DM, Siemens J, Deranian KL, Hayward NJ, Chong JA, Julius D, Moran MM, Fanger CM. 2007. TRPA1 mediates formalin-induced pain. Proc Natl Acad Sci U S A 104:13525-13530. doi:10.1073/pnas.0705924104

Naguib M, Diaz P, Xu JJ, Astruc-Diaz F, Craig S, Vivas-Mejia P, Brown DL. 2008. MDA7: a novel selective agonist for CB_2_ receptors that prevents allodynia in rat neuropathic pain models. Br J Pharmacol. 155:1104-1116. doi: 10.1038/bjp.2008.340.

Song T, Ma X, Ma P, Gu K, Zhao J, Yang Y, Jiang B, Li Y, Wang C. 2018. Administrations of Thalidomide Into the Rostral Ventromedial Medulla Produce Antinociceptive Effects in a Rat Model of Postoperative Pain. J Neurosci Res 96:273-283. doi: 10.1002/jnr.24124